# General Control Functions for Causal Effect Estimation from Instrumental Variables

**Aahlad Puli**
Computer Science
New York University
aahlad@nyu.edu

**Rajesh Ranganath**
Computer Science, Center for Data Science
New York University
rajeshr@cims.nyu.edu

## Abstract

Causal effect estimation relies on separating the variation in the outcome into parts due to the treatment and due to the confounders. To achieve this separation, practitioners often use external sources of randomness that only influence the treatment called instrumental variables (IVs). We study variables constructed from treatment and IV that help estimate effects, called control functions. We characterize general control functions for effect estimation in a meta-identification result. Then, we show that structural assumptions on the treatment process allow the construction of general control functions, thereby guaranteeing identification. To construct general control functions and estimate effects, we develop the general control function method (GCFN). GCFN's first stage called variational decoupling (VDE) constructs general control functions by recovering the residual variation in the treatment given the IV. Using VDE's control function, GCFN's second stage estimates effects via regression. Further, we develop semi-supervised GCFN to construct general control functions using subsets of data that have both IV and confounders observed as supervision; this needs no structural treatment process assumptions. We evaluate GCFN on low and high dimensional simulated data and on recovering the causal effect of slave export on modern community trust [30].

## 1 Introduction

Many disciplines use observational data to estimate causal effects: economics [5], sociology [23], psychology [26], epidemiology [35], and medicine [40]. Estimating causal effects with observational data requires care due to the presence of confounders that influence both treatment and outcome. Observational causal estimators deal with confounders in one of two ways. One, they assume that all confounders are observed; an assumption called *ignorability*. Two, they assume a source of external randomness that has a direct influence only on the treatment. Such a source is called an instrumental variable (IV) [4, 19]. An example is college proximity as an IV to study effects of education [8].

Two common IV-based causal effect estimation methods are the two-stage least-squares method (2SLS) [21, 2, 3] and the traditional control function method (CFN) [19, 45, 43, 12]. Both methods have a common first stage: learn a distribution over the treatment conditioned on the IV. In the second stage, 2SLS regresses the outcome on simulated treatments from the first stage, while CFN's second stage regresses the outcome on the true treatment and the error in the prediction of treatment from the first stage. The prediction error can be used to control for confounding and is thus called a *control function*. Though widely used, both 2SLS and CFN breakdown under certain conditions like, for example, when the outcome depends on multiplicative interactions of treatment and confounders. Further, CFN requires an additional assumption about the correlations between noise and outcome.

We study causal estimation with control functions. To estimate effects, control functions must satisfy ignorability. Our meta-identification result (theorem 1) shows that a control function satisfies ignorability if 1) the control function and IV together reconstruct the treatment, and 2) the confounder and control function together are jointly independent of the IV. We will refer to such control functions

as *general control functions*. Effect estimation in general requires that the treatment has a chance to take any value given the control function; this is called positivity. We show positivity for general control functions holds if the IV can set treatment to any value; we call this a *strong* IV.

Any general control function uniquely determines the effect because it satisfies ignorability and positivity (given a strong IV). Causal identification requires effects to be uniquely determined by the observed data distribution. Thus, building general control functions using observed data guarantees causal identification. As reconstruction and *marginal* independence are properties of the joint distribution over observed data and control function, they can be guaranteed. Guaranteeing *joint* independence requires further assumptions as it involves the *unobserved* confounder. We show that structural assumptions on the treatment process, such as treatment being an additive function of the confounder and IV, help ensure joint independence.

To build general control functions and use them to estimate effects, we develop the general control function method (GCFN). GCFN's first stage, called variational decoupling (VDE), constructs the general control function. VDE is a type of autoencoder where the encoder constructs the control function and the decoder reconstructs treatment from control function and IV, under the constraint that the control function and IV are independent. When VDE is perfectly solved with a decoder that reflects a structural treatment process assumption, like additivity, reconstruction and joint independence are guaranteed. Thus with a strong IV, ignorability and positivity hold which implies identification, and that effect estimation does not require structural assumptions on the *outcome process* like those in 2SLS and CFN. Using VDE's general control function, GCFN's second stage estimates the causal effect. GCFN's second stage can be any method that relies on ignorability like matching/balancing methods [41, 13, 38] and doubly-robust methods [14].

We also consider a setting where a subset of the data has observed confounders that provide ignorability. We develop semi-supervised GCFN to estimate effects in this setting. Semi-supervised GCFN's first stage is an augmented VDE that forces the control function to match the confounder in the subset where it is observed. This augmented VDE helps guarantee joint independence even with a decoder that does not reflect structural treatment process assumptions.

In section 4, we evaluate GCFN's causal effect estimation on simulated data with the outcome, treatment, and IV observed. We demonstrate how GCFN produces correct effect estimates without additional assumptions on the true outcome process, whereas 2SLS, CFN, and DeepIV [18] fail to produce the correct estimate. Further, we show that GCFN performs on par with recently proposed methods DeepGMM [7] and DeepIV [18] on high-dimensional simulations from each respective paper. We also demonstrate that in data with a small subset having observed confounders, semi-supervised GCFN outperforms outcome regression on treatment and confounder within the subset. We also show recovery of the effect of slave export on current societal trust [30].

**Related Work.**   Classical examples of methods that use IVs include the Wald estimator [42], two-stage least-squares method (2SLS) [2, 3, 21] and control function method (CFN) [12, 19, 45, 43]. The Wald estimator assumes constant treatment effect. 2SLS's estimation could be biased when the outcome generating process has multiplicative interactions between treatment and confounders (appendix A.10). Guo and Small [16] proved that under some assumptions, CFN improves upon 2SLS. Beyond these classical estimators, Wooldridge [45] discusses extensions of regression residuals for non-linear models under distributional assumptions about the noise in the treatment process. Hartford et al. [18] developed DeepIV, a deep variant of 2SLS and Singh et al. [39] kernelized the 2SLS algorithm. An alternative to 2SLS is the generalized method of moments (GMM) [17] which solves moment equations implied by the independence of the confounder and the IV. Bennett et al. [7] develop a minimax GMM and use neural networks to specify moment conditions.

Given only an IV, treatment, and outcome, causal effects are not identifiable without further assumptions [6, 25]. Newey [28] and Chetverikov and Wilhelm [10] assume additive outcome processes, where the outcome process is a sum of the causal effect and zero-mean noise; such models are also called separable. Identification in separable models relies on the *completeness* condition [11] which requires the conditional distribution of treatment given IV to sufficiently vary with the IV. Newey [28], Chetverikov and Wilhelm [10] discuss non-parametric estimators under assumptions of monotonicity of the treatment process and shape of causal effects (for eg. $U$-shaped). We focus on the setting where the outcome process *cannot* be represented as a sum of the causal effect and noise, often called a non-separable model [9]. Imbens and Newey [20] showed effect identification in non-separable models when the treatment has a continuous strictly monotonic cumulative distribution

function (CDF) given the IV. Under this same condition, we can guarantee joint independence via a strictly monotonic reconstruction map which means identification holds.

## 1.1 Review of IVs and traditional control function theory

To define the causal effect we use causal graphs [31]. In causal graphs, each variable is represented by a node, and each causal relationship is a directed arrow from the cause to the effect. Causal graphs get transformed by interventions with the do-operator. The shared relationships between the graphs before and after the do-operation make estimation possible. The causal effect of giving a treatment $\mathbf{t} = a$ on an outcome $\mathbf{y}$ is $\mathbb{E}[\mathbf{y} \mid \mathrm{do}(\mathbf{t} = a)]$. The causal

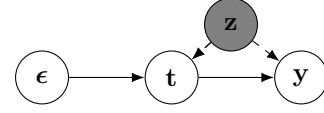

**Figure 1:** Causal graph with hidden confounder $\mathbf{z}$, outcome $\mathbf{y}$, instrument $\epsilon$, treatment $\mathbf{t}$.

graph in Figure 1 describes a broad class of IV problems. The difficulty of causal estimation in this graph stems from the unobserved confounder $\mathbf{z}$. The IV $\epsilon$ helps control for $\mathbf{z}$. Two popular IV-based methods are the two-stage least-squares method (2SLS) and control function method (CFN).

We follow the CFN setup from Guo and Small [16], where the true outcome and treatment processes have additive zero-mean noise called $\boldsymbol{\eta}_{\mathbf{y}}$ and $\boldsymbol{\eta}_{\mathbf{t}}$ that may be correlated due to $\mathbf{z}$:

$$\mathbf{y} = f(\mathbf{t}) + \boldsymbol{\eta}_{\mathbf{y}}, \quad \mathbf{t} = g(\epsilon) + \boldsymbol{\eta}_{\mathbf{t}}. \tag{1}$$

To estimate the causal effect, the CFN method constructs a control function with the regression residual $\mathbf{t} - \hat{g}(\epsilon)$. Then, CFN regresses the outcome $\mathbf{y}$ on the regression residual and the treatment $\mathbf{t}$. The causal effect is the estimate of the function $f(\mathbf{t})$. For this estimate to be valid, the CFN method assumes that $\boldsymbol{\eta}_{\mathbf{t}}, \boldsymbol{\eta}_{\mathbf{y}}$ satisfy the following property for some constant $\rho$, (assumption A4 in [16]):

$$\mathbb{E}[\boldsymbol{\eta}_{\mathbf{y}} \mid \boldsymbol{\eta}_{\mathbf{t}} = \eta] = \rho\eta \tag{2}$$

This property restricts the applicability of the CFN method by limiting how confounders influence the outcome and the treatment. Consider the following additive noise example: $\epsilon, \mathbf{z} \sim \mathcal{N}(0, 1)$, $\mathbf{t} = \mathbf{z} + \epsilon$, $\mathbf{y} \sim \mathcal{N}(\mathbf{t}^2 + \mathbf{z}^2, 1)$, where $\mathcal{N}$ is the standard normal. Here $\eta_y = \mathbf{z}^2$ and $\eta_{\mathbf{t}} = \mathbf{z}$ meaning that $\mathbb{E}[\eta_y \mid \boldsymbol{\eta}_{\mathbf{t}} = \eta] = \eta^2$, violating the assumption in eq. (2). Note that $\mathbb{E}[\mathbf{z}\mathbf{t}^2] = \mathbb{E}[\mathbf{z}\mathbf{z}^2] = 0$, however $\mathbb{E}[\mathbf{t}^2\mathbf{z}^2] > 0$. This means regressing $\mathbf{y}$ on $\mathbf{t}^2$ and $\mathbf{z}$, i.e., with the correct model for $f(\mathbf{t})$, would result in an inflated coefficient of $\mathbf{t}^2$, which is an incorrect causal estimate. Equation (2) is required because some specified function of $\mathbf{t}$ could be correlated with an unspecified function of $\mathbf{z}$, resulting in a biased causal estimate. See appendix A.10 for an example where 2SLS produces biased effect estimates. The assumption in eq. (1) restricts the confounder's influence to be additive on both the treatment and outcome. Further, CFN assumes that the average additive influence the confounder has on the outcome to be a scaled version of the confounder's influence on the treatment (eq. (2)). Such assumptions may not hold in real data. For example, the effect of a medical treatment on patient lifespan is confounded by the patient's current health. This confounder influences the treatment through a human decision process, while it influences the outcome through a physiological process making it unlikely to meet CFN's assumptions.

## 2 Causal Identification with General Control Functions

With a control function that satisfies ignorability and positivity, causal estimation reduces to regression of the outcome on the treatment and the control function. We characterize such control functions:

**Theorem 1. (Meta-identification result for control functions)**
*Let $F(\mathbf{t}, \epsilon, \mathbf{y})$ be the true data distribution. Let control function $\hat{\mathbf{z}}$ be sampled conditionally on $\mathbf{t}, \epsilon$. Let $q(\hat{\mathbf{z}}, \mathbf{t}, \epsilon) = q(\hat{\mathbf{z}} \mid \mathbf{t}, \epsilon)F(\mathbf{t}, \epsilon)$ be the joint distribution over $\hat{\mathbf{z}}, \mathbf{t}, \epsilon$. Further, let $g$ be a deterministic function and $\boldsymbol{\delta}$ be independent noise such that $\mathbf{t} = g(\mathbf{z}, \epsilon, \boldsymbol{\delta})$ and let the implied true joint be $F'(\mathbf{t}, \mathbf{z}, \boldsymbol{\delta})$. Assume the following:*

1. *(A1) $\hat{\mathbf{z}}$ satisfies the **reconstruction** property: $\exists d, \hat{\mathbf{z}}, \mathbf{t}, \epsilon \sim q(\hat{\mathbf{z}}, \mathbf{t}, \epsilon) \implies \mathbf{t} = d(\hat{\mathbf{z}}, \epsilon)$.*

2. *(A2) The IV is **jointly independent** of control function, true confounder, and noise $\boldsymbol{\delta}$: $\epsilon \perp\!\!\!\perp (\mathbf{z}, \hat{\mathbf{z}}, \boldsymbol{\delta})$.*

3. *(A3) **Strong** IV. For any compact $B \subseteq supp(\mathbf{t})$, $\exists c_B$ s.t. a.e. $t \in B$, $F'(\mathbf{t} = t \mid \mathbf{z}, \boldsymbol{\delta}) \geq c_B > 0$.*

*Then, the control function $\hat{\mathbf{z}}$ satisfies ignorability and positivity:*

$$q(\mathbf{y} \mid \mathbf{t} = t, \hat{\mathbf{z}}) = q(\mathbf{y} \mid \mathrm{do}(\mathbf{t} = t), \hat{\mathbf{z}}) \qquad \text{a.e. in } supp(\mathbf{t}) \quad q(\hat{\mathbf{z}}) > 0 \implies q(\mathbf{t} = t \mid \hat{\mathbf{z}}) > 0.$$

*Therefore, the true causal effect is uniquely determined by $q(\hat{\mathbf{z}}, \mathbf{t}, \mathbf{y})$ for almost every $t \in supp(\mathbf{t})$:*

$$\mathbb{E}_{\hat{\mathbf{z}}}[\mathbf{y} \mid \mathbf{t} = t, \hat{\mathbf{z}}] = \mathbb{E}_{\hat{\mathbf{z}}}[\mathbf{y} \mid \mathrm{do}(\mathbf{t} = t), \hat{\mathbf{z}}] = \mathbb{E}[\mathbf{y} \mid \mathrm{do}(\mathbf{t} = t)].$$

Theorem 1 characterizes functions of treatment and IV that satisfy reconstruction (A1) and joint independence (A2) which we call *general control functions*. Positivity of $\mathbf{t}$ w.r.t. the general control function holds under an assumption about the treatment process that the IV is strong (A3). Ignorability and positivity w.r.t. $\hat{\mathbf{z}}$ imply that the true causal effect is uniquely determined as a function of the observed data distribution $q(\hat{\mathbf{z}}, \mathbf{t}, \mathbf{y})$ [1]. If A1 and A2 are satisfied by the observed data distribution $q(\hat{\mathbf{z}}, \mathbf{t}, \mathbf{y}, \boldsymbol{\epsilon})$, the true effect is uniquely determined by the observed data distribution and thus causal identification holds. However, joint independence (A2) relies on the *unobserved* true confounder $\mathbf{z}$. So, theorem 1 is a *meta*-identification result because it does not specify how to guarantee joint independence using $q(\hat{\mathbf{z}}, \mathbf{t}, \boldsymbol{\epsilon})$. In section 2.1, we discuss structural assumptions on the treatment process that instantiate this meta-result and guarantee identification.

Theorem 1 holds for both discrete and continuous $\mathbf{t}$ given that the causal effect exists for all $t \in$ supp$(\mathbf{t})$.[2] While we focus on the causal effect $\mathbb{E}[\mathbf{y} \mid \mathrm{do}(\mathbf{t})]$, theorem 1 guarantees any property of $\mathbf{y} \mid \mathrm{do}(\mathbf{t})$ can be estimated; for e.g. quantile treatment effects. For ease of exposition, we restrict ourselves to treatments of the form $\mathbf{t} = g(\boldsymbol{\epsilon}, \mathbf{z})$, without noise $\boldsymbol{\delta}$. Then, theorem 1 requires only $\boldsymbol{\epsilon} \perp\!\!\!\perp (\mathbf{z}, \hat{\mathbf{z}})$. In appendix A.6, we show $\boldsymbol{\epsilon} \perp\!\!\!\perp (\hat{\mathbf{z}}, \mathbf{z}, \boldsymbol{\delta})$ is guaranteed for more general treatment processes of the form $\mathbf{t} = g(\boldsymbol{\epsilon}, h(\mathbf{z}, \boldsymbol{\delta}))$. Guaranteeing joint independence requires further conditions and is the central challenge in developing two-stage IV-based estimators.

**Why joint independence?** A potential outcome $\mathbf{y_t}$ is the outcome that would be observed if a unit is given treatment $\mathbf{t}$. The potential outcome $\mathbf{y_t}$ follows the distribution of $\mathbf{y}$ under the *do* operator and only depends on the true confounder $\mathbf{z}$. For ignorability with respect to $\hat{\mathbf{z}}$, we need $\mathbf{y_t}$ to be independent of $\mathbf{t}$, given $\hat{\mathbf{z}}$. By reconstruction, given $\hat{\mathbf{z}}$, $\mathbf{t}$ is purely a function of $\boldsymbol{\epsilon}$. This means ignorability with respect to the control function $\hat{\mathbf{z}}$ requires that the true confounder and IV be independent given the control function. Therefore, ignorability requires $\mathbf{z} \perp\!\!\!\perp \boldsymbol{\epsilon} \mid \hat{\mathbf{z}}$. Further, conditional independence $\hat{\mathbf{z}} \perp\!\!\!\perp \boldsymbol{\epsilon} \mid \mathbf{z}$ implies positivity of $\mathbf{t}$ w.r.t $\hat{\mathbf{z}}$ if $\boldsymbol{\epsilon}$ is strong. Joint independence $\boldsymbol{\epsilon} \perp\!\!\!\perp (\mathbf{z}, \hat{\mathbf{z}})$ implies both the conditional independencies above.

The causal graph fig. 1 with $\mathbf{y}$ marginalized out can be represented with two sources of randomness one from the unobserved confounder $\mathbf{z}$ and one from the IV $\boldsymbol{\epsilon}$; the extra randomness in $\mathbf{t}$ denoted as $\boldsymbol{\delta}$ can be absorbed into $\mathbf{z}$. In this setup, the treatment and control function are deterministic functions of the unobserved confounder and IV. With only two sources of randomness, joint independence means the control function $\hat{\mathbf{z}}$ needs to only be a function of the true unobserved confounder $\mathbf{z}$. When $\hat{\mathbf{z}}$ is a stochastic function of the treatment and IV, joint independence holds if $\hat{\mathbf{z}}$ determines $\mathbf{z}$ while $\hat{\mathbf{z}} \perp\!\!\!\perp \boldsymbol{\epsilon}$.

As $\hat{\mathbf{z}}$ and $\boldsymbol{\epsilon}$ are observed, we can guarantee $\hat{\mathbf{z}} \perp\!\!\!\perp \boldsymbol{\epsilon}$. The marginal independence $\mathbf{z} \perp\!\!\!\perp \boldsymbol{\epsilon}$ holds by definition of an IV. However, even both marginal independencies $\hat{\mathbf{z}} \perp\!\!\!\perp \boldsymbol{\epsilon}$ and $\mathbf{z} \perp\!\!\!\perp \boldsymbol{\epsilon}$ together do not imply joint independence $\boldsymbol{\epsilon} \perp\!\!\!\perp (\hat{\mathbf{z}}, \mathbf{z})$. This means a control function $\hat{\mathbf{z}}$ that satisfies the reconstruction property and marginal independence $\hat{\mathbf{z}} \perp\!\!\!\perp \boldsymbol{\epsilon}$ may fail to yield ignorability. In appendix A.4, we build an example of a deterministic almost everywhere invertible function of two independent variables $\mathbf{c} = f(\mathbf{a}, \mathbf{b})$ such that $\mathbf{c} \perp\!\!\!\perp \mathbf{a}$ and $\mathbf{c} \perp\!\!\!\perp \mathbf{b}$ and yet, joint independence $(\mathbf{c}, \mathbf{b}) \not\!\perp\!\!\!\perp \mathbf{a}$ is violated. As $\mathbf{z}$ is unobserved, achieving joint independence requires further assumptions. Next, we discuss how structural assumptions on the true treatment process can help guarantee joint independence.

## 2.1 Guaranteeing joint independence for identification

We show structural treatment process assumptions help guarantee joint independence by relating it to $q(\hat{\mathbf{z}}, \mathbf{t}, \boldsymbol{\epsilon})$ and thus giving identification. Joint independence can be guaranteed (via marginal independence) if the reconstruction map $d(\hat{\mathbf{z}}, \boldsymbol{\epsilon})$ (A1, theorem 1) reflects the functional structure of the treatment process. As an example, consider an additive treatment process $\mathbf{t} = \mathbf{z} + g(\boldsymbol{\epsilon})$. If the reconstruction map is $d(\hat{\mathbf{z}}, \boldsymbol{\epsilon}) = h'(\hat{\mathbf{z}}) + g'(\boldsymbol{\epsilon})$ and $\boldsymbol{\epsilon} \perp\!\!\!\perp \hat{\mathbf{z}}$, joint independence holds. To see this, note

$$h'(\hat{\mathbf{z}}) - \mathbb{E}_{\hat{\mathbf{z}}}[h'(\hat{\mathbf{z}})] = \mathbf{t} - \mathbb{E}[\mathbf{t} \mid \boldsymbol{\epsilon}] = \mathbf{z} - \mathbb{E}_{\mathbf{z}}[\mathbf{z}] \implies \exists \text{ constant } c, h'(\hat{\mathbf{z}}) = \mathbf{z} + c, \quad (3)$$

meaning $h'(\hat{\mathbf{z}})$ determines $\mathbf{z}$. By $\hat{\mathbf{z}} \perp\!\!\!\perp \boldsymbol{\epsilon}$, it holds that $q(\hat{\mathbf{z}}, \mathbf{z} \mid \boldsymbol{\epsilon}) = q(\hat{\mathbf{z}}, h'(\hat{\mathbf{z}}) - c \mid \boldsymbol{\epsilon}) = q(\hat{\mathbf{z}}, \mathbf{z})$. Thus, leveraging the functional structure of the treatment process helps guarantee joint independence by relating it to $q(\hat{\mathbf{z}}, \mathbf{t}, \boldsymbol{\epsilon})$, via $\hat{\mathbf{z}} \perp\!\!\!\perp \boldsymbol{\epsilon}$. Assuming treatment gets generated from other *known* invertible functions, such as multiplication $\mathbf{t} = h(\mathbf{z}) * g(\boldsymbol{\epsilon})$, also leads to joint independence. Imbens and Newey [20] proved effect identification when the treatment is a continuous strictly monotonic function of the confounder; these conditions helps guarantee joint independence (see appendix A.7). For more general treatments of the form $\mathbf{t} = g(\boldsymbol{\epsilon}, h(\mathbf{z}, \boldsymbol{\delta}))$ the structural assumptions from above can only guarantee $(h(\mathbf{z}, \boldsymbol{\delta}), \hat{\mathbf{z}}) \perp\!\!\!\perp \boldsymbol{\epsilon}$; see appendix A.5 for general additive treatments: $\mathbf{t} = h(\mathbf{z}, \boldsymbol{\delta}) + g(\boldsymbol{\epsilon})$. However, we show in appendix A.6 that for such general treatment processes $(h(\mathbf{z}, \boldsymbol{\delta}), \hat{\mathbf{z}}) \perp\!\!\!\perp \boldsymbol{\epsilon} \implies (\mathbf{z}, \hat{\mathbf{z}}, \boldsymbol{\delta}) \perp\!\!\!\perp \boldsymbol{\epsilon}$ which, together with reconstruction, implies ignorability(theorem 1). In summary, under certain structural assumptions, general control functions exist ($\hat{\mathbf{z}} = \mathbf{z}$ for example) and can be built using only properties of the observed data distribution $q(\hat{\mathbf{z}}, \mathbf{t}, \boldsymbol{\epsilon})$. This guarantees identification. In section 3, we develop practical algorithms to build general control functions.

## 2.2 Comparison of identification with general control functions to existing work

Traditional CFN theory [16] relies on the assumption that the treatment process is additive; recall $\mathbf{t} = g(\boldsymbol{\epsilon}) + \boldsymbol{\eta}_t$ from section 1.1 and $\boldsymbol{\eta}_t$ is correlated with outcome noise due to $\mathbf{z}$. Beyond this additivity assumption, traditional CFN theory further assumes 1) the outcome process is additive, like in eq. (1), 2) the noise $\boldsymbol{\eta}_y$ in the outcome process is independent of the IV, 3) linear noise relationship between $\boldsymbol{\eta}_t, \boldsymbol{\eta}_y$, like in eq. (2), and 4) (relevance) the treatment effect function and IV are correlated [16]. When the treatment process is additive, joint independence can be guaranteed as a property of the distribution $q(\hat{\mathbf{z}}, \mathbf{t}, \boldsymbol{\epsilon})$, via $\hat{\mathbf{z}} \perp\!\!\!\perp \boldsymbol{\epsilon}$; see section 2.1. Then, identification with general control functions requires a strong IV. While it allows structural outcome process assumptions (like 3) can be relaxed, a strong IV needs more than the two IV properties, independence with confounder and relevance. However, domain expertise helps reason about strong IVs; for example, can college proximity influence a student's decision to go to college regardless of skill? If yes, college proximity is a strong IV. We compare against other identification conditions (like 2SLS and [20]) in appendix A.8.

# 3 The General Control Function Method (GCFN)

GCFN constructs a general control function and estimates effects with it. GCFN has two stages. The first stage constructs a general control function as the code of an autoencoder. The second stage builds a model from the control function and the treatment to the outcome and estimates effects.

**Variational Decoupling** We construct the control function $\hat{\mathbf{z}}$ as a stochastic function of the treatment $\mathbf{t}$ and the IV $\boldsymbol{\epsilon}$; with parameter $\theta$, the estimator is $q_\theta(\hat{\mathbf{z}} \mid \mathbf{t}, \boldsymbol{\epsilon})$. First, to guarantee the reconstruction property (A1 in theorem 1), the control function and the IV must determine treatment, implying that with parameter $\phi$, $p_\phi(\mathbf{t} \mid \hat{\mathbf{z}} = \hat{z}, \boldsymbol{\epsilon})$ should be maximized for $\hat{z} \sim q(\hat{\mathbf{z}} \mid \mathbf{t}, \boldsymbol{\epsilon})$. Together, these form the parts of an autoencoder where a control function is sampled conditioned on the treatment and IV, while the treatment is reconstructed from the same control function and IV. Second, to guarantee marginal independence, we force the control function to be independent of the IV: $\hat{\mathbf{z}} \perp\!\!\!\perp \boldsymbol{\epsilon}$. Let the true data distribution be $F(\mathbf{t}, \boldsymbol{\epsilon})$ and $\mathbf{I}$ denote mutual information. Putting the two parts together, we define a constrained optimization to construct $\hat{\mathbf{z}}$, called variational decoupling (VDE):

$$\text{(VDE)} \quad \max_{\theta, \phi} \mathbb{E}_{F(\mathbf{t}, \boldsymbol{\epsilon})} \mathbb{E}_{q_\theta(\hat{\mathbf{z}} \mid \mathbf{t}, \boldsymbol{\epsilon})} \log p_\phi(\mathbf{t} \mid \hat{\mathbf{z}}, \boldsymbol{\epsilon}) \quad s.t \quad \mathbf{I}_\theta(\hat{\mathbf{z}}; \boldsymbol{\epsilon}) = 0. \tag{4}$$

Recall from section 2.1 that with a reconstruction map $d(\hat{\mathbf{z}}, \boldsymbol{\epsilon})$ (from A1 in theorem 1) that reflects the functional structure of the treatment, marginal independence $\hat{\mathbf{z}} \perp\!\!\!\perp \boldsymbol{\epsilon}$ implies joint independence. To model such a map, VDE's decoder, $p_\phi(\mathbf{t} \mid \hat{\mathbf{z}} = \hat{z}, \boldsymbol{\epsilon})$ reflects the same functional structure. For example, with an additive treatment process the decoder would be parametrized as $\log p_\phi(\mathbf{t} \mid \hat{\mathbf{z}}, \boldsymbol{\epsilon}) \propto -(\mathbf{t} - h'_\phi(\hat{\mathbf{z}}) - g'_\phi(\boldsymbol{\epsilon}))/\sigma_\phi^2$; $\sigma_\phi$ allows for a point-mass distribution $p_\phi$ at optimum of VDE. In summary, beyond the observed treatment and IV, VDE takes a specification of the functional structure of the treatment process as input which informs the structure of the decoder.

VDE is converted to an unconstrained optimization problem by absorbing the independence constraint into the optimization via the Lagrange multipliers trick with $\lambda > 0$,

$$\max_{\theta, \phi} \mathbb{E}_{F(\mathbf{t}, \boldsymbol{\epsilon})} \mathbb{E}_{q_\theta(\hat{\mathbf{z}} \mid \mathbf{t}, \boldsymbol{\epsilon})} \log p_\phi(\mathbf{t} \mid \hat{\mathbf{z}}, \boldsymbol{\epsilon}) - \lambda \mathbf{I}_\theta(\hat{\mathbf{z}}; \boldsymbol{\epsilon}). \tag{5}$$

Estimation of the mutual information requires $q_\theta(\hat{\mathbf{z}} \mid \boldsymbol{\epsilon})$. Instead, we lower bound the negative mutual information by introducing an auxiliary distribution $r_\nu(\hat{\mathbf{z}})$. This yields a tractable objective:

$$\max_{\theta,\phi,\nu} \mathbb{E}_{F(\mathbf{t},\boldsymbol{\epsilon})}\left[(1+\lambda)\mathbb{E}_{q_\theta(\hat{\mathbf{z}} \mid \mathbf{t},\boldsymbol{\epsilon})} \log p_\phi(\mathbf{t} \mid \hat{\mathbf{z}},\boldsymbol{\epsilon}) - \lambda \mathrm{KL}\left(q_\theta(\hat{\mathbf{z}} \mid \mathbf{t},\boldsymbol{\epsilon}) \parallel r_\nu(\hat{\mathbf{z}})\right)\right]. \tag{6}$$

A full derivation can be found in Appendix A.2. The lower bound is tight when the auxiliary distribution $r_\nu(\hat{\mathbf{z}}) = q_\theta(\hat{\mathbf{z}})$. For example, when $q_\theta(\hat{\mathbf{z}} \mid \mathbf{t},\boldsymbol{\epsilon})$ is categorical, optimizing eq. (6) with a categorical $r_\nu(\hat{\mathbf{z}})$ makes the lower bound tight. The parameters $\theta,\phi,\nu$ can be learned via stochastic optimization. VDE can be adapted to use covariates by conditioning on the covariates as needed.

**Outcome Modeling.** VDE provides a general control function $\hat{\mathbf{z}}$ and its marginal distribution $q_\theta(\hat{\mathbf{z}})$. If the IV is strong, $\hat{\mathbf{z}}$ satisfies ignorability and positivity and the causal effect can be estimated by regressing the outcome on the control function and the treatment. Other effect estimation methods like matching/balancing methods [41, 13, 38] and doubly-robust methods [14] can be used. This regression is GCFN's second stage, called the outcome stage. We formalize this outcome stage as a maximum-likelihood problem and learn a model with parameters $\beta$ under the true data distribution $F(\mathbf{y},\mathbf{t},\boldsymbol{\epsilon})$ and the general control function distribution $q_\theta(\hat{\mathbf{z}} \mid \mathbf{t},\boldsymbol{\epsilon})$:

$$\arg\max_\beta \mathbb{E}_{F(\mathbf{y},\mathbf{t},\boldsymbol{\epsilon})} \mathbb{E}_{q_\theta(\hat{\mathbf{z}} \mid \mathbf{t},\boldsymbol{\epsilon})} \log p_\beta\left(\mathbf{y} \mid \hat{\mathbf{z}}, \mathbf{t}\right). \tag{7}$$

**Semi-Supervised GCFN.** The explicit optimization to learn the control function in VDE makes it simple to take advantage of datapoints where both the confounder and IV are observed by forcing the control function to predict the observed confounder. Let $\mathbf{m}$ be an missingness indicator variable that is 1 when the true confounder $\mathbf{z}$ is observed and 0 otherwise. Let the joint distribution be $F(\mathbf{t},\boldsymbol{\epsilon},\mathbf{m},\mathbf{z})$ and $\zeta$ be a scaling hyperparameter parameter. Then the augmented VDE stage in semi-supervised GCFN, with $\kappa = \lambda/(1+\lambda)$, is

$$\max_{\theta,\phi,\nu} \mathop{\mathbb{E}}_{F(\mathbf{t},\boldsymbol{\epsilon},\mathbf{m},\mathbf{z})}\left[\mathop{\mathbb{E}}_{q_\theta(\hat{\mathbf{z}} \mid \mathbf{t},\boldsymbol{\epsilon})} \log p_\phi(\mathbf{t} \mid \hat{\mathbf{z}},\boldsymbol{\epsilon}) - \kappa\mathrm{KL}\left(q_\theta(\hat{\mathbf{z}} \mid \mathbf{t},\boldsymbol{\epsilon}) \parallel r_\nu(\hat{\mathbf{z}})\right) + \zeta\mathbf{m}\log q_\theta(\hat{\mathbf{z}} = \mathbf{z} \mid \mathbf{t},\boldsymbol{\epsilon}).\right] \tag{8}$$

The added term $\log q_\theta(\hat{\mathbf{z}} = \mathbf{z} \mid \mathbf{t},\boldsymbol{\epsilon})$ encourages the control function to place all of its mass on the observed confounder value. When the control function places all of its mass on the confounder, the control function is determined by value of the confounder. Together with the fact that the confounder is independent of the IV, this implies the control function, confounder pair is jointly independent of the instrument. Therefore, given enough datapoints with the confounder and IV observed, joint independence can be guaranteed without treatment assumptions like in section 2.1. The second stage of semi-supervised GCFN uses the outcome regression in eq. (7) to estimate effects.

## 3.1 Error bounds for GCFN's estimated effects

An imperfectly estimated general control function may violate the conditional independence $\mathbf{z} \perp\!\!\!\perp \boldsymbol{\epsilon} \mid \hat{\mathbf{z}}$ which is required for ignorability. If ignorability does not hold, estimated effects are biased. First, assuming an additive treatment process, we bound the expected bias in causal effects using quantities optimized during training in VDE, specifically reconstruction error and dependence of $\hat{\mathbf{z}}$ on $\boldsymbol{\epsilon}$:

**Theorem 2.** *Assume an additive treatment process $\mathbf{t} = \mathbf{z} + g(\boldsymbol{\epsilon})$ where $g$ is an $L_g$-Lipschitz function, and $\mathbb{E}_{F(\mathbf{z})}\mathbf{z} = 0$. Let $\mathbb{E}[\mathbf{y} \mid \mathbf{t} = t, \mathbf{z} = z] = f(t,z)$ be an $L$-Lipschitz function in $z$ for any $t$. Further,*

1. *let reconstruction error be non-zero but bounded $\mathbb{E}_{q(\mathbf{t},\hat{\mathbf{z}},\boldsymbol{\epsilon})}(\mathbf{t} - \hat{\mathbf{z}} - g'(\boldsymbol{\epsilon}))^2 \leq \delta$. Assume that $g'$ is also $L_g$-Lipschitz. Further, let $\mathbb{E}_{q(\hat{\mathbf{z}})}\hat{\mathbf{z}} = 0$, and $\mathbb{E}_{q(\hat{\mathbf{z}})}|\hat{\mathbf{z}}| < \infty$.*

2. *Assume $\boldsymbol{\epsilon} \not\!\perp\!\!\!\perp \hat{\mathbf{z}}$ and let the dependence be bounded: $\max_{\hat{z}} \mathcal{W}_1\left(q(\boldsymbol{\epsilon} \mid \hat{\mathbf{z}} = \hat{z}) \parallel F(\boldsymbol{\epsilon})\right) \leq \gamma$.*

*With the estimated and true causal effects as $\hat{\tau}(t) = \mathbb{E}_{\hat{\mathbf{z}}}f(t,\hat{\mathbf{z}})$ and $\tau(t) = E_\mathbf{z}f(t,\mathbf{z})$ respectively,*

$$\mathbb{E}_{F(\mathbf{t})}|\hat{\tau}(\mathbf{t}) - \tau(\mathbf{t})| \leq L\sqrt{\delta + 4\gamma L_g \mathbb{E}_{q(\hat{\mathbf{z}})}|\hat{\mathbf{z}}|}.$$

See appendix A.9.1 for the proof. Second, in theorem 3 in appendix A.9.2, we prove a general error bound for GCFN that depends on the residual confounding that $\hat{\mathbf{z}}$ does not control for, measured as the conditional mutual information $\mathbf{I}(\mathbf{z}; \mathbf{t} \mid \hat{\mathbf{z}})$. When $\mathbf{I}(\mathbf{z}; \mathbf{t} \mid \hat{\mathbf{z}}) > 0$, ignorability may not hold and estimated effects are biased. Assuming positivity and a sufficiently concentrated $\mathbf{z} \mid \hat{\mathbf{z}}$, we prove in theorem 3 that $\mathbf{I}(\mathbf{z}; \mathbf{t} \mid \hat{\mathbf{z}})$ controls average absolute error in effects. This error is tempered by the smoothness of outcome as a function of the confounder $\mathbf{z}$. This bound also accounts for errors due to poor estimation of $\mathbb{E}[\mathbf{y} \mid \mathbf{t}, \hat{\mathbf{z}}]$ in low density regions of $q(\mathbf{t}, \hat{\mathbf{z}})$ which may occur when $\hat{\mathbf{z}} \not\!\perp\!\!\!\perp \boldsymbol{\epsilon}$.

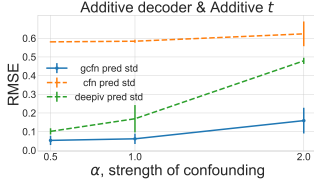
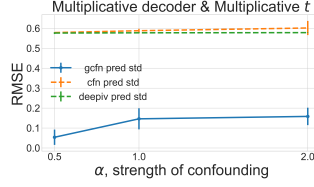
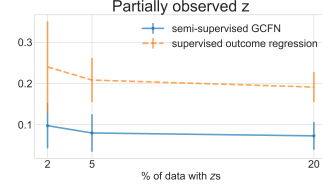

**Figure 2:** GCFN obtains better effect estimates than CFN and DeepIV when the *additive outcome process* assumption is violated.

**Figure 3:** GCFN produces better effect estimates than CFN and DeepIV when the *additive treatment* process assumption is violated.

**Figure 4:** Mean RMSE of causal effects of the GCFN-predicted causal effects versus percentages of samples with **z** observed.

## 4 Experiments

We evaluate GCFN on simulated data, where the true causal effects are known and show that GCFN corrects for confounding and estimates causal effects better than CFN, 2SLS, and a 2SLS variant, DeepIV [18]. We then evaluate GCFN on high-dimensional data using simulations from DeepIV [18] and DeepGMM [7]. Then, we estimate the effect of slave export on community trust [30] and compare GCFN's estimate to the effect reported in [30].

**Experimental details** For GCFN, we let the control function $\hat{\mathbf{z}}$ be a categorical variable. The encoder in VDE, $f_\theta$, is a 2-hidden-layer neural network $f_\theta$, which parametrizes a categorical likelihood $q_\theta(\hat{\mathbf{z}} = i \mid \mathbf{t} = t, \boldsymbol{\epsilon} = \epsilon) \propto \exp(f_\theta(t, \epsilon, i))$. The decoder is also a 2-hidden-layer network; the reconstructed likelihood of $\mathbf{t}$ is different for different experiments. In all experiments, the hidden layers in both encoder and decoder networks have 100 units and use ReLU activations. The outcome model is also a 2-hidden-layer neural network with ReLU activations. For the simulated data, the hidden layers in the outcome model have 50 hidden units. In estimating the effect of slave export, the hidden layers in the outcome model have only 10 hidden units; larger width resulted in overfitting. Unless specified otherwise, we train on 5000 samples with a batch size of 500 for optimizing both VDE and the outcome model for 100 epochs with Adam [22]. In section 4.1 and section 4.2, we evaluate effect estimates on a subset of the support of the treatment distribution where the most mass lies: 200 equally spaced treatment values in $[-1, 1]$. We defer other details to appendix B.

All hyperparameters for VDE, except the mutual-information coefficient $\kappa = \lambda/(1 + \lambda)$, and the outcome-stage were found by evaluating the respective objectives on a held-out validation set. In our experiments, we found that setting $\kappa$ between $0.1 - 0.4$ worked best. GCFN's performance was only mildly sensitive to changing $\kappa$ within this range. However, one can tune $\kappa$ further by choosing the one which gives the control function $\hat{\mathbf{z}}_\kappa$ that results in the largest expected outcome likelihood on a heldout set. This procedure relies on VDE and outcome objectives reaching optimum if and only if $\hat{\mathbf{z}}$ satisfies perfect reconstruction and marginal independence. See appendix B.1 for further details.

### 4.1 Simulations with specific decoder structure

We compare GCFN's performance against 2SLS, CFN and DeepIV and show that GCFN outperforms these methods when the functional properties of the treatment process are known. We consider two settings with continuous outcome, treatment, and confounders where the assumptions of 2SLS and CFN fail: 1) with an additive treatment process and a multiplicative outcome process and 2) with a multiplicative treatment process and an additive outcome process. For both settings, the causal effect is the same $\mathbb{E}[\mathbf{y} \mid \mathrm{do}(\mathbf{t} = t)] = t$. The control function $\hat{\mathbf{z}}$ is set to have 50 categories. We report results for the mutual information coefficient $\kappa = \lambda/_{1+\lambda} = 0.1$. We consider 3 different strengths of confounding as captured by the parameter $\alpha \in [0.5, 1.0, 2.0]$.

**Multiplicative outcome & Additive treatment** With $\mathcal{N}$ as the normal distribution, we generate $\mathbf{z}, \boldsymbol{\epsilon} \sim \mathcal{N}(0, 1)$, $\mathbf{t} = (\mathbf{z} + \boldsymbol{\epsilon})/\sqrt{2}$, $\mathbf{y} \sim \mathcal{N}(\mathbf{t} + \alpha \mathbf{t}^2 \mathbf{z}, 0.1)$, wher $\alpha$ controls confounding; larger magnitude of $\alpha$ means more confounding. The generation process above violates the linear noise relation assumption, $\mathbb{E}[\boldsymbol{\eta_y} | \boldsymbol{\eta_t}] \propto \boldsymbol{\eta_t}$, that CFN requires [16]. GCFN, on the other hand, does not require this assumption. In this experiment, VDE has an additive decoder which specifies a Gaussian reconstruction likelihood: $\mathbf{t} \sim \mathcal{N}(h'_\phi(\hat{\mathbf{z}}) + g'_\phi(\boldsymbol{\epsilon}), 1)$. In Figure 2, we compare GCFN to CFN

and DeepIV, and show that GCFN produces the best causal effect estimates. Unlike the others, GCFN can adjust for confounding when the outcome process is not additive. Averaged over all $\alpha$s, GCFN outperforms the baselines with an RMSE of $\mathbf{0.09 \pm 0.06}$ compared to CFN's $\mathbf{0.58 \pm 0.01}$, 2SLS's $\mathbf{0.55 \pm 0.58}$, and DeepIV's $\mathbf{0.25 \pm 0.17}$.

**Multiplicative treatment & Additive outcome.** For this simulation, we generate data as follows: $\mathbf{z}, \boldsymbol{\epsilon} \sim \mathcal{N}(0, 1)$, $\mathbf{t} = \mathbf{z}\boldsymbol{\epsilon}$, $\mathbf{y} \sim \mathcal{N}(\mathbf{t} + \alpha\mathbf{z}, 0.1)$. In this experiment, VDE has a *multiplicative* decoder which specifies a gaussian reconstruction likelihood with $\mathbf{t} = \mathcal{N}(h'_\phi(\hat{\mathbf{z}})g'_\phi(\boldsymbol{\epsilon}), 1)$. The 2SLS method uses a linear model $\mathbf{t} = \beta\boldsymbol{\epsilon} + \boldsymbol{\eta_t}$ which will correctly estimate $\mathbb{E}[\mathbf{t} \mid \boldsymbol{\epsilon}] = 0$ in our generation process. Figure 3 shows that GCFN out-performs CFN and DeepIV and is robust to different strengths of confounding ($\alpha \in \{0.5, 1, 2\}$). Averaged over all $\alpha$s, GCFN outperforms the baselines with an RMSE of $\mathbf{0.13 \pm 0.08}$ compared to CFN's $\mathbf{0.58 \pm 0.02}$, 2SLS's $\mathbf{0.55 \pm 0.56}$, and DeepIV's $\mathbf{0.58 \pm 0.01}$. We omit 2SLS from fig. 2 because it performs strictly worse than DeepIV, its deep variant. DeepIV gives effect-estimates that are close to 0. We justify this in appendix A.11.

## 4.2 GCFN with confounders observed on a subset

In this experiment, we demonstrate that semi-supervised GCFN does not need outcome or treatment process assumptions if the confounder $\mathbf{z}$ is observed on a subset of the data. Let $\rho$ be the fraction with $\mathbf{z}$ observed and $\mathcal{B}$ be the Bernoulli distribution. We generate a mask $\mathbf{m} \sim \mathcal{B}(\rho)$ and data $\boldsymbol{\epsilon}, \mathbf{z} \sim \mathcal{N}(0, 1)$, $\mathbf{t} = \boldsymbol{\epsilon}\mathbf{z}$, $\mathbf{y} \sim \mathcal{N}(\mathbf{t} + \mathbf{t}\mathbf{z}, 0.1)$. Let $\mathbf{z}' = \mathbf{z} * \mathbf{m}$. We observe $(\mathbf{y}, \mathbf{t}, \boldsymbol{\epsilon}, \mathbf{z}', \mathbf{m})$. The structurally unrestricted decoder uses a categorical reconstruction likelihood: $p_\phi(\mathbf{t} = j \mid \hat{\mathbf{z}} = z, \boldsymbol{\epsilon} = \epsilon) \propto \exp(g_\phi(z, \epsilon, j))$. The treatment $\mathbf{t}$ is discretized into 50 bins. The intervals $[-\infty, -3.5]$ and $[3.5, \infty]$ correspond to one bin each and the interval $[-3.5, 3.5]$ is split into 48 equally-sized bins. This suffices because few samples fall outside $[-3.5, 3.5]$. For semi-supervised GCFN, VDE's objective has an additional term defined on the samples with observed $\mathbf{z}$'s (eq. (8)). The confounder $\mathbf{z}$ is split into bins the same way as the treatment. The additional term for the $i^{th}$ sample is the categorical log-likelihood of the observed $(t_i, \epsilon_i, z_i)$ with respect to the encoder-specified distribution: $q(\hat{\mathbf{z}} = z_i \mid \mathbf{t} = t_i, \boldsymbol{\epsilon} = \epsilon_i) \propto \exp(f_\theta(t_i, \epsilon_i, z_i))$. We set the scaling $\zeta$ on this additional term to be 0.5. We report results for $\kappa = 0.1$. For other $\kappa \in \{0.2, 0.3\}$, results were similar or better.

We compare semi-supervised GCFN against regression with the same outcome model as the baseline, trained only on samples with the confounder observed. We estimated this "supervised" baseline in the same manner as the outcome stage of GCFN. Figure 4 plots the RMSE of the predicted causal effects vs. percentage of samples with observed $\mathbf{z}$'s in fig. 4. If the data has $2\%$ or more samples with the confounder observed, GCFN estimates effects better than the supervised baseline.

## 4.3 GCFN on high-dimensional Covariates

In this experiment, we evaluate GCFN on a non-linear simulation given in Hartford et al. [18] to demonstrate that DeepIV improves upon 2SLS. Their generation models the effect of price ($\mathbf{t}$) on sales ($\mathbf{y}$), given customer covariates ($\mathbf{x}$, MNIST image), and time $s$; they use fuel price as an IV. The outcome is generated using the label of the MNIST image, which denotes customer price sensitivity. The data generation process for $\mathbf{t}$ is additive in IV and confounder. Following this, we use the same additive decoder in VDE as in section 4.1, but with time $s$ as an additional input. We give further experimental details and Hartford et al. [18]'s data generating process in appendix B.3.

We report effect MSE on a fixed out-of-sample set (OOS). We compare against Hartford et al. [18]'s reported results for two sample sizes, $10,000$ and $20,000$. DeepIV's reported results exclude a few large effect MSE outliers; we do not exclude such errors for GCFN. We report GCFN's performance over 10 seeds. Overall, GCFN performed on par or better than DeepIV. First, we report GCFN's effect MSE with $\kappa = 0.2$. For $10,000$ samples, GCFN produced effect MSEs that ranged in $[\mathbf{0.30 - 0.42}]$, better than DeepIV's reported range of around $[\mathbf{0.30 - 0.50}]$ (which is almost twice as large). For $20,000$ samples, GCFN's effect MSE range improved to $[\mathbf{0.25 - 0.40}]$ while DeepIV reported a performance of around $[\mathbf{0.25 - 0.45}]$. For both sample sizes, we note that $\kappa = 0.1, 0.3$ gave similar results. To see this, for $20,000$ samples, averaged over 10 seeds, GCFN achieved a mean effect MSE of $\mathbf{0.305}$ or better for any $\kappa \in \{0.1, 0.2, 0.3\}$, beating DeepIV's $\mathbf{0.32}$.

### 4.4 GCFN on high-dimensional IVs

In this experiment, we evaluate GCFN on data with a high-dimensional IV. Bennett et al. [7] use the following data generating process to demonstrate DeepGMM [7] improves upon existing methods: $\epsilon \sim \mathcal{U}[-3,3]$ $\quad \mathbf{z} \sim \mathcal{N}(0,1)$ $\quad \mathbf{t} \sim \mathcal{N}(\mathbf{z}+\epsilon, 0.1)$ $\quad \mathbf{y} = \mathcal{N}(|\mathbf{t}| + \mathbf{z}, 0.1)$. However, the scalar $\epsilon$ is not directly observed. Instead, $\epsilon$ is mapped to a digit $\{0, \ldots, 9\}$ and a corresponding MNIST image $\epsilon_M$ is given as the IV. To estimate effects well with such an IV, any method must learn to label the MNIST image. In this setting, VDE's encoder and decoder both take an embedding $\ell_\gamma(\epsilon_M) \in \mathbb{R}^{10}$ as input. The embedding $\ell_\gamma$ is trained in VDE along with the encoder and decoder. Respecting the additive treatment process, we specify an additive decoder.

We ran GCFN with 10 different random seeds and report results for $\kappa = 0.3$, chosen based on mean test outcome MSE ($0.136 \pm 0.008$). GCFN performs competitively with an effect MSE of $\mathbf{0.077 \pm 0.022}$ compared to DeepGMM's $\mathbf{0.07 \pm 0.02}$ and DeepIV's $\mathbf{0.11 \pm 0.00}$, both as reported in [7]. Effect MSE for $\kappa \in \{0.2, 0.4\}$ were similar and within standard error of DeepGMM's performance. See appendix B.4 for further experimental details and results.

### 4.5 The Effect of Slave Export on Trust

We demonstrate the recovery of the causal effect of slave export on the trust in the community [30]. Nunn and Wantchekon [30] pooled surveys and historical records to get sub-ethnicity and tribe level data from the period of slave trade. The data was used to study the long-term effects of slave-trade, measured in the 2005 Afrobarometer survey. We predict the effect of the treatment $\mathbf{t} =$**ln(1 + slave-export/area)** on the outcome of interest, $\mathbf{y} =$**trust in neighbors**. The dataset has 6932 samples with 59 features. After filtering out missing values, we preprocessed 46 covariates and IV to have mean 0 and maximum 1, and the treatment $\mathbf{t}$ to lie in $[0, 2]$. The authors claim that the distance to sea cannot causally affect how individuals trust each other, but it affects the chance of coming in contact with colonial slave-traders and being shipped to the Americas, making it an IV. They control for urbanization, fixed effects for sophistication, political hierarchies beyond community, integration with the rail network, contact with European explorers, and missions during colonial rule.

For this experiment, VDE's decoder $g_\phi$ specifies a categorical reconstruction likelihood as $p_\phi(\mathbf{t} = i \mid \hat{\mathbf{z}} = z, \epsilon = \epsilon) \propto \exp(g_\phi(z, \epsilon, i))$. Each category of the treatment corresponds to one of 50 equally-sized bins in the interval $[0, 2]$. Nunn and Wantchekon [30] use a linear model for the outcome $\mathbf{y}$ and use the distance to sea as an IV for each community. We also use a partially linear model $\mathbf{y} = \beta \mathbf{t} + h_\theta(\hat{\mathbf{z}})$ so that the effect we recover is of comparable nature to the effect reported in the paper. The outcome network $h_\theta$ has 2 layers with 10 hidden units each and ReLUs.

Averaged over 4 mutual information coefficients $\kappa$ and 5 random seeds, GCFN's estimate of $\beta$ was $\mathbf{-0.21 \pm 0.04}$ compared with $\mathbf{-0.27 \pm 0.10}$, as reported by Nunn and Wantchekon [30].

## 5 Discussion and Future

In this paper, we characterize general control functions for causal estimation. General control functions allow for effect estimation without structural outcome process assumptions like 2SLS or CFN. The key challenge in building general control functions is ensuring joint independence between the IV and the control function and (unobserved) true confounder. Joint independence can be guaranteed via structural treatment process assumptions, like additivity or monotonicity. We develop the general control function method (GCFN) to build general control functions and estimate effects with them. Further, we develop semi-supervised GCFN which uses confounders observed on a subset of the data to construct general control functions without treatment process assumptions. Finally, we consider imperfect estimation of the general control function and bound average error in effects using quantities optimized in VDE.

**Tradeoffs with assumptions.** In causal estimation, parametric assumptions can be traded-off with assumptions of strength of IV or positivity. Consider a setting where $\epsilon$ is binary. For every possible confounder value, only two values of the treatment are observed. Thus it is impossible to estimate a quadratic function of $\mathbf{t}$ for each fixed value of the confounder. This means $\mathbf{y} \mid \mathbf{t}$ is not identified without strong assumptions like linearity in $\mathbf{t}$. Incorporating outcome properties, like the conditional independence $\mathbf{y} \perp\!\!\!\perp \epsilon \mid \mathbf{t}, \mathbf{z}$, into control function estimation would be a fruitful direction.

## Broader Impact

Our work applies to causal inference where strong IVs are available to help adjust for confounding, such as in problems in healthcare and economics. We assess the impact of our work in the context of these fields. In general, loosening functional assumptions like GCFN does, helps estimate effects better. Better effect estimates help improve planning patient treatment and understanding policy impact. However, the strong IV assumption may not hold for all demographics. If this occurs, demographics for which the assumption holds will have better quality effect estimates than for demographics where the assumption does not hold. This could mean that certain demographics receive better care in hospitals or have implemented policy be more impactful on them. Such issues could be characterized by evaluating the positivity of treatment with respect to the constructed control function in different demographics.

## Acknowledgements

The authors were partly supported by NIH/NHLBI Award R01HL148248, and by NSF Award 1922658 NRT-HDR: FUTURE Foundations, Translation, and Responsibility for Data Science. The authors would like to thank Xintian Han and the reviewers for thoughtful feedback.

## Footnotes

[1]We also require that $\mathbb{E}_{\hat{\mathbf{z}}}\mathbb{E}[\mathbf{y} \mid \mathrm{do}(\mathbf{t}), \hat{\mathbf{z}}]$ exists. This is guaranteed if the causal effect $\mathbb{E}[\mathbf{y} \mid \mathrm{do}(\mathbf{t})] = \mathbb{E}_{\mathbf{z}}\mathbb{E}[\mathbf{y} \mid \mathbf{t}, \mathbf{z}]$ exists as ignorability holds w.r.t. $\hat{\mathbf{z}}$: $\mathbb{E}_{\hat{\mathbf{z}}}\mathbb{E}[\mathbf{y} \mid \mathrm{do}(\mathbf{t}), \hat{\mathbf{z}}] = \mathbb{E}_{\hat{\mathbf{z}}}\mathbb{E}_{q(\mathbf{z} \mid \hat{\mathbf{z}})}\mathbb{E}[\mathbf{y} \mid \mathbf{t}, \mathbf{z}] = \mathbb{E}_{\mathbf{z}}\mathbb{E}[\mathbf{y} \mid \mathbf{t}, \mathbf{z}].$

[2]Effects for certain treatments can be identified even without the strong IV assumption (A3): for any compact subset $B \subseteq$ supp$(\mathbf{t})$ such that $\forall t \in B$, $F'(\mathbf{t} = t \mid \mathbf{z}, \boldsymbol{\delta}) \geq c_B > 0$, effects can be estimated for all $t \in B$.

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
