[Supplementary Material]



# A   Theoretical Details and Proofs

**Notation**   We use the expectation operator in different contexts in the proof. $\mathbb{E}_q$ denotes expectation with respect to the density $q$ and $\mathbb{E}_{\mathbf{z}}$ denotes expectation with respect to the density of the random variable $\mathbf{z}$. When the density function or the random variable are clear from the context, we drop the subscript and use $\mathbb{E}$.

## A.1   The general IV causal graph with covariates/observed confounders

**Figure 5:** Causal graph with hidden confounder $\mathbf{z}$, outcome $\mathbf{y}$, IV $\boldsymbol{\epsilon}$, treatment $\mathbf{t}$ and covariates $\mathbf{x}$.

Figure 5 is the general version of the IV problem where the instrumental variable property holds true after conditioning on $\mathbf{x}$. This is sometimes called a conditional instrument. All our proofs and results carry over to the situation with covariates after conditioning all estimables and distributions on $\mathbf{x}$. VDE in this setting with covariates is re-written as:

$$\max_{\theta,\phi} \mathbb{E}_{F(\mathbf{t},\boldsymbol{\epsilon},\mathbf{x})}\mathbb{E}_{q_\theta(\hat{\mathbf{z}} \mid \mathbf{t},\boldsymbol{\epsilon},\mathbf{x})} \log p_\phi(\mathbf{t} \mid \hat{\mathbf{z}}, \boldsymbol{\epsilon}, \mathbf{x}) - \lambda \mathbf{I}_\theta(\hat{\mathbf{z}}; \boldsymbol{\epsilon} \mid \mathbf{x}) \tag{9}$$

## A.2   Mutual Information lower bound

Here, we show the full derivation of the lower bound for negative mutual-information. We derive the lower bound for the general case where there are both observed and unobserved confounders. A simple lower bound can be obtained by using $\mathbf{H}(\hat{\mathbf{z}} \mid \boldsymbol{\epsilon}, \mathbf{x}) \geq \mathbf{H}(\hat{\mathbf{z}} \mid \boldsymbol{\epsilon}, \mathbf{t}, \mathbf{x})$, but this cannot be made tight unless $\boldsymbol{\epsilon}$ completely determines $\mathbf{t}$. Therefore, we cannot guarantee independence unless the data at hand is not confounded. Instead we introduce two auxiliary distributions $r_\nu(\hat{\mathbf{z}} \mid \mathbf{x})$ and $p_\phi(\mathbf{t} \mid \boldsymbol{\epsilon}, \hat{\mathbf{z}}, \mathbf{x})$, following the work in variational inference [34, 1, 36, 24] and causal inference [33].

We let $F(\mathbf{t}, \mathbf{x}, \boldsymbol{\epsilon}, \mathbf{y})$ be the true data distribution and $q_\theta(\hat{\mathbf{z}} \mid \mathbf{t}, \boldsymbol{\epsilon}, \mathbf{x} = x)$ be the control function distribution. We overload notation and also use $q_\theta$ to refer to any distribution that involves operations with $q_\theta(\hat{\mathbf{z}} \mid \mathbf{t}, \boldsymbol{\epsilon}, \mathbf{x} = x)$. We use $\overset{c}{=}$ to denote that the LHS and RHS are equal up to constants that are ignored during optimization. In the following, both $\mathbf{H}(\mathbf{t}, \boldsymbol{\epsilon} \mid \mathbf{x} = x)$, $\mathbf{H}(\boldsymbol{\epsilon} \mid \mathbf{x} = x)$ are constants with respect to the parameters of interest $\phi, \theta, \nu$ and we will drop them from the lower bound when encountered. For a given $\mathbf{x} = x$, we lower-bound the negative instantaneous conditional mutual information:

$$-\lambda \mathbf{I}(\hat{\mathbf{z}}; \boldsymbol{\epsilon} \mid x) = -\lambda \mathrm{KL}\left(q_\theta(\hat{\mathbf{z}}, \boldsymbol{\epsilon} \mid x) \parallel q_\theta(\hat{\mathbf{z}} \mid x) F(\boldsymbol{\epsilon} \mid x)\right)$$

$$= -\lambda \left[\mathbb{E}_{q_\theta(\boldsymbol{\epsilon},\hat{\mathbf{z}} \mid x)}\left[\log q_\theta(\boldsymbol{\epsilon} \mid \hat{\mathbf{z}}, x) - \log F(\boldsymbol{\epsilon} \mid x)\right]\right]$$

$$= -\lambda \left[\mathbb{E}_{q_\theta(\boldsymbol{\epsilon},\hat{\mathbf{z}} \mid x)}\left[\log q_\theta(\boldsymbol{\epsilon} \mid \hat{\mathbf{z}}, x)\right] + \mathbf{H}(\boldsymbol{\epsilon} \mid x)\right]$$

$$\overset{c}{=} -\lambda \left[\mathbb{E}_{q_\theta(\boldsymbol{\epsilon},\hat{\mathbf{z}} \mid x)}\left[\mathrm{KL}\left(q_\theta(\hat{\mathbf{z}} \mid x) \parallel q_\theta(\hat{\mathbf{z}} \mid x)\right) + \mathrm{KL}\left(q_\theta(\mathbf{t} \mid \boldsymbol{\epsilon}, \hat{\mathbf{z}}, x) \parallel q_\theta(\mathbf{t} \mid \boldsymbol{\epsilon}, \hat{\mathbf{z}}, x)\right) + \log q_\theta(\boldsymbol{\epsilon} \mid \hat{\mathbf{z}}, x)\right]\right]$$

$$\geq -\lambda \left[\mathbb{E}_{q_\theta(\boldsymbol{\epsilon},\hat{\mathbf{z}} \mid x)}\left[\mathrm{KL}\left(q_\theta(\hat{\mathbf{z}} \mid x) \parallel r_\nu(\hat{\mathbf{z}} \mid x)\right) + \mathrm{KL}\left(q_\theta(\mathbf{t} \mid \boldsymbol{\epsilon}, \hat{\mathbf{z}}, x) \parallel p_\phi(\mathbf{t} \mid \boldsymbol{\epsilon}, \hat{\mathbf{z}}, x)\right) + \log q_\theta(\boldsymbol{\epsilon} \mid \hat{\mathbf{z}}, x)\right]\right]$$

$$= -\lambda \left[\mathbb{E}_{q_\theta(\boldsymbol{\epsilon},\hat{\mathbf{z}} \mid x)}\left[\log\left[q_\theta(\hat{\mathbf{z}}, \boldsymbol{\epsilon} \mid x)\right] + \mathbb{E}_{q_\theta(\mathbf{t} \mid \boldsymbol{\epsilon},\hat{\mathbf{z}},x)} \log q_\theta(\mathbf{t} \mid \boldsymbol{\epsilon}, \hat{\mathbf{z}}, x)\right.\right.$$
$$\left.\left. - \mathbb{E}_{q_\theta(\hat{\mathbf{z}} \mid x)} \log r_\nu(\hat{\mathbf{z}} \mid x) - \mathbb{E}_{q_\theta(\mathbf{t},\boldsymbol{\epsilon},\hat{\mathbf{z}} \mid x)} \log p_\phi(\mathbf{t} \mid \boldsymbol{\epsilon}, \hat{\mathbf{z}}, x)\right]\right]$$

$$= -\lambda \left[\mathbb{E}_{q_\theta(\boldsymbol{\epsilon},\hat{\mathbf{z}},\mathbf{t} \mid x)} \log\left[q_\theta(\hat{\mathbf{z}}, \boldsymbol{\epsilon}, \mathbf{t} \mid x)\right] - \mathbb{E}_{q_\theta(\hat{\mathbf{z}} \mid x)} \log r_\nu(\hat{\mathbf{z}} \mid x) - \mathbb{E}_{q_\theta(\mathbf{t},\boldsymbol{\epsilon},\hat{\mathbf{z}} \mid x)} \log p_\phi(\mathbf{t} \mid \boldsymbol{\epsilon}, \hat{\mathbf{z}}, x)\right]$$

$$= -\lambda \left[\mathbb{E}_{F(\mathbf{t},\boldsymbol{\epsilon} \mid x)}\mathbb{E}_{q_\theta(\hat{\mathbf{z}} \mid \boldsymbol{\epsilon},\mathbf{t},x)} \log\left[q_\theta(\hat{\mathbf{z}} \mid \mathbf{t}, \boldsymbol{\epsilon}, x) - \log p_\phi(\mathbf{t} \mid \boldsymbol{\epsilon}, \hat{\mathbf{z}}, x)\right] - \mathbf{H}(\mathbf{t}, \boldsymbol{\epsilon} \mid x)\right.$$
$$\left. - \mathbb{E}_{q_\theta(\hat{\mathbf{z}} \mid x)} \log r_\nu(\hat{\mathbf{z}} \mid x)\right]$$

$$\overset{c}{=} -\lambda \mathbb{E}_{F(\mathbf{t},\boldsymbol{\epsilon} \mid x)}\left[\mathrm{KL}\left(q_\theta(\hat{\mathbf{z}} \mid \mathbf{t}, \boldsymbol{\epsilon}, x) \parallel r_\nu(\hat{\mathbf{z}} \mid x)\right) - \mathbb{E}_{q_\theta(\hat{\mathbf{z}} \mid \boldsymbol{\epsilon},\mathbf{t},x)} \log p_\phi(\mathbf{t} \mid \boldsymbol{\epsilon}, \hat{\mathbf{z}}, x)\right],$$

where the hidden term $-\lambda\left[\mathbf{H}(\boldsymbol{\epsilon} \mid \mathbf{x} = x) - \mathbf{H}(\mathbf{t}, \boldsymbol{\epsilon} \mid \mathbf{x} = x)\right]$ is a constant for a given instance of the problem. We do not need access to the distribution $\mathbf{t}, \hat{\mathbf{z}}, \boldsymbol{\epsilon} \mid \mathbf{x} = x$ because the information that we lower bounded, $\mathbf{I}(\hat{\mathbf{z}}; \boldsymbol{\epsilon} \mid \mathbf{x} = x)$, is averaged over $\mathbf{x} = x$ in our objective. Recall that $p_\phi(\mathbf{t} \mid \boldsymbol{\epsilon}, \hat{\mathbf{z}}, \mathbf{x} = x)$ is the reconstruction term in VDE. This lower bound is tight when the introduced KL terms are 0, which occurs when $r_\nu(\hat{\mathbf{z}} \mid \mathbf{x} = x) = q_\theta(\hat{\mathbf{z}} \mid \mathbf{x} = x)$ and $p_\phi(\mathbf{t} \mid \boldsymbol{\epsilon}, \mathbf{x} = x, \hat{\mathbf{z}}) = q_\theta(\mathbf{t} \mid \boldsymbol{\epsilon}, \mathbf{x} = x, \hat{\mathbf{z}})$. This means that if the models $p_\phi, r_\nu$ are rich enough, the gap between the lower bound and mutual information can be optimized to be zero. The second term $\mathbb{E}_{q_\theta(\hat{\mathbf{z}} \mid \boldsymbol{\epsilon}, \mathbf{t}, \mathbf{x}=x)} \log p_\phi(\mathbf{t} \mid \boldsymbol{\epsilon}, \hat{\mathbf{z}}, \mathbf{x} = x)$ is the same as the reconstruction likelihood. Thus substituting the lower bound into the full objective with given covariates gives

$$\mathbb{E}_{F(\mathbf{t}, \boldsymbol{\epsilon}, \mathbf{x})}\left[(1 + \lambda)\mathbb{E}_{q_\theta(\hat{\mathbf{z}} \mid \mathbf{t}, \boldsymbol{\epsilon}, \mathbf{x})} \log p_\phi(\mathbf{t} \mid \boldsymbol{\epsilon}, \hat{\mathbf{z}}, \mathbf{x}) - \lambda \mathrm{KL}\left(q_\theta(\hat{\mathbf{z}} \mid \mathbf{t}, \boldsymbol{\epsilon}, \mathbf{x}) \,\|\, r_\nu(\hat{\mathbf{z}} \mid \mathbf{x})\right)\right]$$

**Optimization for variational decoupling (VDE).** The VDE optimization involves the expectations of distributions with parameters with respect to a distribution that also has parameters. For distributions that are not being integrated against, we can move the gradient inside the expectation. For distributions that are integrated against, score-function methods provide a general tool to compute stochastic gradients; Glasserman [15], Williams [44], Ranganath et al. [32], Mnih and Gregor [27]. In our experiments, we let the control function be a categorical variable. This allows us to marginalize out the control function and compute the gradient.

## A.3 Proof of Theorem 1

**Theorem 1.** *(Meta-identification result for general control functions)*
*Let $F(\mathbf{t}, \boldsymbol{\epsilon}, \mathbf{y})$ be the true data distribution. Let control function $\hat{\mathbf{z}}$ be sampled conditionally on $\mathbf{t}, \boldsymbol{\epsilon}$. Let $q(\hat{\mathbf{z}}, \mathbf{t}, \boldsymbol{\epsilon}) = q(\hat{\mathbf{z}} \mid \mathbf{t}, \boldsymbol{\epsilon})F(\mathbf{t}, \boldsymbol{\epsilon})$ be the joint distribution over $\hat{\mathbf{z}}, \mathbf{t}, \boldsymbol{\epsilon}$. Further, let $g$ be a deterministic function and $\boldsymbol{\delta}$ be independent noise such that $\mathbf{t} = g(\mathbf{z}, \boldsymbol{\epsilon}, \boldsymbol{\delta})$ and let the implied true joint be $F'(\mathbf{t}, \mathbf{z}, \boldsymbol{\delta})$. Assume the following:*

1. *(A1) $\hat{\mathbf{z}}$ satisfies the **reconstruction** property: $\exists d, \hat{\mathbf{z}}, \mathbf{t}, \boldsymbol{\epsilon} \sim q(\hat{\mathbf{z}}, \mathbf{t}, \boldsymbol{\epsilon}) \implies \mathbf{t} = d(\hat{\mathbf{z}}, \boldsymbol{\epsilon})$.*

2. *(A2) The IV is **jointly independent** of control function, true confounder, and noise $\boldsymbol{\delta}$: $\boldsymbol{\epsilon} \perp\!\!\!\perp (\mathbf{z}, \hat{\mathbf{z}}, \boldsymbol{\delta})$.*

3. *(A3) **Strong** IV. For any compact $B \subseteq supp(\mathbf{t})$, $\exists c_B$ s.t. a.e. $t \in B$, $F'(\mathbf{t} = t \mid \mathbf{z}, \boldsymbol{\delta}) \geq c_B > 0$.*

*Then, the control function $\hat{\mathbf{z}}$ satisfies ignorability and positivity:*

$$q(\mathbf{y} \mid \mathbf{t} = t, \hat{\mathbf{z}}) = q(\mathbf{y} \mid \mathrm{do}(\mathbf{t} = t), \hat{\mathbf{z}}) \qquad \text{a.e. in } supp(\mathbf{t}) \quad q(\hat{\mathbf{z}}) > 0 \implies q(\mathbf{t} = t \mid \hat{\mathbf{z}}) > 0.$$

*Therefore, the true causal effect is uniquely determined by $q(\hat{\mathbf{z}}, \mathbf{t}, \mathbf{y})$ for almost every $t \in supp(\mathbf{t})$:*

$$\mathbb{E}_{\hat{\mathbf{z}}}[\mathbf{y} \mid \mathbf{t} = t, \hat{\mathbf{z}}] = \mathbb{E}_{\hat{\mathbf{z}}}[\mathbf{y} \mid \mathrm{do}(\mathbf{t} = t), \hat{\mathbf{z}}] = \mathbb{E}[\mathbf{y} \mid \mathrm{do}(\mathbf{t} = t)].$$

We prove this for the setting without covariates. The proof adapts to the setting with covariates (observed confounders) by conditioning all terms on them.

*Proof.* (Theorem 1) The proof shows that reconstruction (A1) and joint independence (A2) together imply ignorability, and strong IV (A3) together with the joint independence (A2) imply positivity.

**Ignorability.** To establish ignorability we need to show that $\mathbf{y}_t \perp\!\!\!\perp \mathbf{t} \mid \hat{\mathbf{z}}$ where $\mathbf{y}_t$ is the potential outcome for a unit when the treatment given is $\mathbf{t} = t$. The outcome $\mathbf{y}$ is constructed from the potential outcomes by indexing the one $\mathbf{y}_{t^*}$ corresponding to the observed treatment $\mathbf{t} = t^*$.

By assumption A2, we have the joint independence $\boldsymbol{\epsilon} \perp\!\!\!\perp (\mathbf{z}, \hat{\mathbf{z}})$ which implies

$$\boldsymbol{\epsilon} \perp\!\!\!\perp (\mathbf{z}, \hat{\mathbf{z}}) \implies \boldsymbol{\epsilon} \perp\!\!\!\perp \mathbf{z} \mid \hat{\mathbf{z}} = \hat{z} \quad \forall \hat{z} \in \mathrm{supp}(\hat{\mathbf{z}}).$$

Note that by the reconstruction property (from assumption A1) $\mathbf{t} = d(\hat{\mathbf{z}}, \boldsymbol{\epsilon})$. So given $\hat{\mathbf{z}}$, $\mathbf{t}$ is purely a function of $\boldsymbol{\epsilon}$. Thus, given $\hat{\mathbf{z}}$, $\mathbf{t}$ satisfies the same conditional independence as $\boldsymbol{\epsilon}$: $\boldsymbol{\epsilon} \perp\!\!\!\perp \mathbf{z} \mid \hat{\mathbf{z}}$. Using this, we have

$$\boldsymbol{\epsilon} \perp\!\!\!\perp \mathbf{z} \mid \hat{\mathbf{z}} \implies d(\hat{\mathbf{z}}, \boldsymbol{\epsilon}) \perp\!\!\!\perp \mathbf{z} \mid \hat{\mathbf{z}} \implies \mathbf{t} \perp\!\!\!\perp \mathbf{z} \mid \hat{\mathbf{z}}.$$

The potential outcome $\mathbf{y}_t$ depends only on $\mathbf{z}$ and some noise $\boldsymbol{\eta}$ that is jointly independent of all other variables. This means for some function $m_t$ such that $\mathbf{y}_t = m_t(\mathbf{z}, \boldsymbol{\eta})$.

$$\mathbf{t} \perp\!\!\!\perp \mathbf{z} \mid \hat{\mathbf{z}} \implies \mathbf{t} \perp\!\!\!\perp m_t(\mathbf{z}, \boldsymbol{\eta}) \mid \hat{\mathbf{z}} \implies \mathbf{t} \perp\!\!\!\perp \mathbf{y}_t \mid \hat{\mathbf{z}}.$$

This shows ignorability.

**Strength of IV and Positivity.** Positivity means that for almost every $t \in \text{supp}(\mathbf{t})$,

$$q(\hat{\mathbf{z}}) > 0, \implies q(\mathbf{t} = t \mid \hat{\mathbf{z}}) > 0.$$

We start with $q(\mathbf{t} \mid \hat{\mathbf{z}})$ and expand it as an integral over the full joint.

$$
\begin{aligned}
q(\mathbf{t} \mid \hat{\mathbf{z}}) &= \int q(\mathbf{t} \mid \mathbf{z} = z, \hat{\mathbf{z}}, \boldsymbol{\epsilon} = \epsilon, \boldsymbol{\delta} = \delta, \mathbf{t}) q(\boldsymbol{\epsilon} = \epsilon \mid \mathbf{z} = z, \hat{\mathbf{z}}, \boldsymbol{\delta} = \delta) q(\mathbf{z} = z, \boldsymbol{\delta} = \delta \mid \hat{\mathbf{z}}) dz d\delta d\epsilon \\
&= \int q(\mathbf{t} \mid \mathbf{z} = z, \boldsymbol{\epsilon} = \epsilon, \boldsymbol{\delta} = \delta) q(\boldsymbol{\epsilon} = \epsilon \mid \mathbf{z} = z, \hat{\mathbf{z}}, \boldsymbol{\delta} = \delta) q(\mathbf{z} = z, \boldsymbol{\delta} = \delta \mid \hat{\mathbf{z}}) dz d\delta d\epsilon \\
&\quad \{\text{by } \mathbf{t} = g(\mathbf{z}, \boldsymbol{\epsilon}, \boldsymbol{\delta})\} \\
&= \int q(\mathbf{t} \mid \mathbf{z} = z, \boldsymbol{\epsilon} = \epsilon, \boldsymbol{\delta} = \delta) q(\boldsymbol{\epsilon} = \epsilon \mid \mathbf{z} = z, \boldsymbol{\delta} = \delta) q(\mathbf{z} = z, \boldsymbol{\delta} = \delta \mid \hat{\mathbf{z}}) dz d\delta d\epsilon \\
&\quad \{\text{by A2: } \boldsymbol{\epsilon} \perp\!\!\!\perp (\mathbf{z}, \hat{\mathbf{z}}, \boldsymbol{\delta})\} \\
&= \int \left[ \int q(\mathbf{t} \mid \mathbf{z} = z, \boldsymbol{\epsilon} = \epsilon, \boldsymbol{\delta} = \delta) q(\boldsymbol{\epsilon} = \epsilon \mid \mathbf{z} = z, \boldsymbol{\delta} = \delta) d\epsilon \right] q(\mathbf{z} = z, \boldsymbol{\delta} = \delta \mid \hat{\mathbf{z}}) dz d\delta \\
&= \int F'(\mathbf{t} \mid \mathbf{z} = z, \boldsymbol{\delta} = \delta) q(\mathbf{z} = z, \boldsymbol{\delta} = \delta \mid \hat{\mathbf{z}}) dz d\delta
\end{aligned}
$$

(10)

Note that $q(\mathbf{z} = z, \boldsymbol{\delta} = \delta \mid \hat{\mathbf{z}})$ is a valid density over $(\mathbf{z} = z, \boldsymbol{\delta} = \delta)$ [3]. Under assumption A3, for any compact set $B \subseteq \text{supp}(\mathbf{t})$ and for almost every $t \in B$,

$$
\begin{aligned}
q(\mathbf{t} = t \mid \hat{\mathbf{z}}) &= \int F'(\mathbf{t} = t \mid \mathbf{z} = z, \boldsymbol{\delta} = \delta) q(\mathbf{z} = z, \boldsymbol{\delta} = \delta \mid \hat{\mathbf{z}}) dz d\delta \\
&\geq c_B \int q(\mathbf{z} = z, \boldsymbol{\delta} = \delta \mid \hat{\mathbf{z}}) dz d\delta \\
&= c_B > 0
\end{aligned}
$$

(11)

However, almost every $t \in \text{supp}(\mathbf{t})$ is contained in some compact subset $B \subseteq \text{supp}(\mathbf{t})$. Thus, eq. (11) holds for almost every $t \in \text{supp}(\mathbf{t})$, meaning that positivity is satisfied.

**Computing the causal effect.** Given ignorability and positivity, the true causal effect (a.e. in $\text{supp}(\mathbf{t})$) is determined as a property of the distribution $q(\hat{\mathbf{z}}, \mathbf{t}, \mathbf{y})$ as follows:

$$\mathbb{E}_{q(\hat{\mathbf{z}})} \mathbb{E}[\mathbf{y} \mid \hat{\mathbf{z}}, \mathbf{t} = t] = \mathbb{E}_{q(\hat{\mathbf{z}})} \mathbb{E}[\mathbf{y} \mid \hat{\mathbf{z}}, \text{do}(\mathbf{t} = t)] = \mathbb{E}[\mathbf{y} \mid \text{do}(\mathbf{t} = t)]$$

$\square$

**Assumptions for continuous $\mathbf{t}$.** When $\mathbf{t}$ has non-zero density rather than non-zero probability given the general control function, the true expected outcome being continuous everywhere as a function of the treatment is a sufficient condition for the causal effect estimation for almost all treatment values.

### A.4 Marginal Independence does not imply joint independence

Here, we build an example of a function of two independent variables $\mathbf{a}, \mathbf{b}$ that is marginally independent of both. Let $1_e$ be one if $e$ is true and zero if not,

$$
\begin{aligned}
\mathbf{a}, \mathbf{b} &\sim \text{uniform}(0, 1), \\
\mathbf{c}(\mathbf{a}, \mathbf{b}) &= 1_{\mathbf{a}+\mathbf{b}>1}(\mathbf{a} + \mathbf{b} - 1) + 1_{\mathbf{a}+\mathbf{b}\leq 1}(\mathbf{a} + \mathbf{b}).
\end{aligned}
$$

First, $\mathbf{c}$ is marginally a uniform variable.[4] The distribution $\mathbf{c} \mid \mathbf{a} = x$ can be obtained by translating the distribution of $\mathbf{b}$ up by $x$, then translating the part greater than one down to zero, meaning $\mathbf{c} \mid \mathbf{a}$ is

uniformly distributed. Thus $p(\mathbf{c} \mid \mathbf{a}) = p(\mathbf{c})$ meaning $\mathbf{c} \perp\!\!\!\perp \mathbf{a}$. However, $\mathbf{c}$ is a deterministic function of $\mathbf{a}$ and $\mathbf{b}$. Therefore, while $\mathbf{c} \mid \mathbf{a}$ is uniformly distributed, $\mathbf{c} \mid (\mathbf{a}, \mathbf{b})$ is a dirac-delta distribution, meaning $p(\mathbf{c} \mid \mathbf{a}, \mathbf{b}) \neq p(\mathbf{c} \mid \mathbf{a})$ implying $\mathbf{c} \not\perp\!\!\!\perp \mathbf{a} \mid \mathbf{b}$. Note that $\mathbf{b}$ can be constructed back from $\mathbf{c}, \mathbf{a}$ up to measure-zero as $\mathbf{b} = \mathbf{c} - \mathbf{a}$ if $\mathbf{c} > \mathbf{a}$ and $\mathbf{b} = \mathbf{c} - \mathbf{a} + 1$ if $\mathbf{c} \leq \mathbf{a}$; i.e., $\mathbf{c}$ is almost everywhere invertible for each fixed $\mathbf{a} = a$.

This construction with uniform random variables can be generalized to other continuous distributions by inverse transform sampling. Any marginal density of $\mathbf{a}, \mathbf{b}$ can be bijectively mapped to a uniform density over $[0, 1]$. Then $\mathbf{c}$ can be computed as above and then $\mathbf{a}, \mathbf{b}, \mathbf{c}$ can be bijectively mapped back; $\mathbf{c}$ could be mapped back with the CDF of $\mathbf{b}$. Conditional dependence is unaffected by bijective transformations and therefore the issue remains. Similar constructions exist with discrete random variables. In general, assumptions on the true data generating process will be needed to ensure joint independence.

## A.5    From additive treatment processes to joint independence

Consider treatment processes of the form $\mathbf{t} = h(\mathbf{z}, \boldsymbol{\delta}) + g(\boldsymbol{\epsilon})$. Let the reconstruction map be additive:

$$\mathbf{t} = h'(\hat{\mathbf{z}}) + g'(\boldsymbol{\epsilon}).$$

Consider the random variable $\mathbf{t} - \mathbb{E}[\mathbf{t} \mid \boldsymbol{\epsilon}]$ which is sampled as follows: $\boldsymbol{\epsilon} \sim h(\boldsymbol{\epsilon}), \mathbf{z} \sim h(\mathbf{z}), \boldsymbol{\delta} \sim h(\boldsymbol{\delta})$ and $\mathbf{t} - \mathbb{E}[\mathbf{t} \mid \boldsymbol{\epsilon}] = h(\mathbf{z}, \boldsymbol{\delta}) + g(\boldsymbol{\epsilon}) - \mathbb{E}_{\mathbf{z}, \boldsymbol{\delta}}[h(\mathbf{z}, \boldsymbol{\delta}) + g(\boldsymbol{\epsilon})]$. We show that $h'(\hat{\mathbf{z}})$ determines $h(\mathbf{z}, \boldsymbol{\delta})$ by expressing the random variable $\mathbf{t} - \mathbb{E}[\mathbf{t} \mid \boldsymbol{\epsilon}]$ in terms of $\mathbf{z}, \boldsymbol{\delta}$ and $\hat{\mathbf{z}}$

$$h'(\hat{\mathbf{z}}) - \mathbb{E}_{\hat{\mathbf{z}}}[h'(\hat{\mathbf{z}})] = \mathbf{t} - \mathbb{E}[\mathbf{t} \mid \boldsymbol{\epsilon}] = h(\mathbf{z}, \boldsymbol{\delta}) - \mathbb{E}_{\mathbf{z}, \boldsymbol{\delta}}[h(\mathbf{z}, \boldsymbol{\delta})].$$

Therefore for some constant $c$, $h'(\hat{\mathbf{z}}) = h(\mathbf{z}, \boldsymbol{\delta}) + c$. By the independence, $\hat{\mathbf{z}} \perp\!\!\!\perp \boldsymbol{\epsilon}$, we have

$$q(\hat{\mathbf{z}}, h(\mathbf{z}, \boldsymbol{\delta}) \mid \boldsymbol{\epsilon}) = q(\hat{\mathbf{z}}, h'(\hat{\mathbf{z}}) - c \mid \boldsymbol{\epsilon}) = q(\hat{\mathbf{z}}, h'(\hat{\mathbf{z}}) - c) = q(\hat{\mathbf{z}}, h(\mathbf{z}, \boldsymbol{\delta})).$$

Thus we have $(\hat{\mathbf{z}}, h(\mathbf{z}, \boldsymbol{\delta})) \perp\!\!\!\perp \boldsymbol{\epsilon}$. See lemma 1 for the proof that $(\hat{\mathbf{z}}, h(\mathbf{z}, \boldsymbol{\delta})) \perp\!\!\!\perp \boldsymbol{\epsilon}$ implies the joint independence $\boldsymbol{\epsilon} \perp\!\!\!\perp (\hat{\mathbf{z}}, \mathbf{z}, \boldsymbol{\delta})$ for any treatment process $\mathbf{t} = g(\boldsymbol{\epsilon}, h(\mathbf{z}, \boldsymbol{\delta}))$, including $\mathbf{t} = g(\boldsymbol{\epsilon}) + h(\mathbf{z}, \boldsymbol{\delta})$.

## A.6    Joint independence treatments of the form $\mathbf{t} = g(\boldsymbol{\epsilon}, h(\mathbf{z}, \boldsymbol{\delta}))$

General control functions for treatments of the form $\mathbf{t} = g(\boldsymbol{\epsilon}, h(\mathbf{z}, \boldsymbol{\delta}))$, unlike $\mathbf{t} = g(\boldsymbol{\epsilon}, \mathbf{z})$, require a stronger joint independence $\boldsymbol{\epsilon} \perp\!\!\!\perp (\mathbf{z}, \hat{\mathbf{z}}, \boldsymbol{\delta})$ to guarantee ignorability (A2, theorem 1). The structural assumptions — that helped guarantee $\boldsymbol{\epsilon} \perp\!\!\!\perp (\mathbf{z}, \hat{\mathbf{z}})$ above — can guarantee $\boldsymbol{\epsilon} \perp\!\!\!\perp (h(\mathbf{z}, \boldsymbol{\delta}), \hat{\mathbf{z}})$. Here, we show that $\boldsymbol{\epsilon} \perp\!\!\!\perp (h(\mathbf{z}, \boldsymbol{\delta}), \hat{\mathbf{z}}) \implies \boldsymbol{\epsilon} \perp\!\!\!\perp (\mathbf{z}, \hat{\mathbf{z}}, \boldsymbol{\delta})$ in such settings.

**Lemma 1.** *Consider treatment process* $\mathbf{t} = g(\boldsymbol{\epsilon}, h(\mathbf{z}, \boldsymbol{\delta}))$ *and the joint independence* $(\hat{\mathbf{z}}, h(\mathbf{z}, \boldsymbol{\delta})) \perp\!\!\!\perp \boldsymbol{\epsilon}$ *holds. Then, if* $\hat{\mathbf{z}} = e(\mathbf{t}, \boldsymbol{\epsilon})$*, the joint independence* $(\hat{\mathbf{z}}, \mathbf{z}, \boldsymbol{\delta}) \perp\!\!\!\perp \boldsymbol{\epsilon}$ *holds.*

*Proof.* We begin by showing $q(\hat{\mathbf{z}} \mid \mathbf{z}, \boldsymbol{\epsilon}, \boldsymbol{\delta}) = q(\hat{\mathbf{z}} \mid h(\mathbf{z}, \boldsymbol{\delta}))$:

$$
\begin{aligned}
q(\hat{\mathbf{z}} \mid \mathbf{z}, \boldsymbol{\epsilon}, \boldsymbol{\delta}) &= \int q(\hat{\mathbf{z}} \mid \mathbf{z}, \boldsymbol{\epsilon}, \mathbf{t} = t, \boldsymbol{\delta}) q(\mathbf{t} = t \mid \boldsymbol{\epsilon}, \mathbf{z}, \boldsymbol{\delta}) dt \quad \{\text{full joint expansion}\} \\
&= \int q(\hat{\mathbf{z}} \mid \boldsymbol{\epsilon}, \mathbf{t} = t) q(\mathbf{t} = t \mid \boldsymbol{\epsilon}, \mathbf{z}, \boldsymbol{\delta}) dt \quad \{\hat{\mathbf{z}} \perp\!\!\!\perp (\mathbf{z}, \boldsymbol{\delta}) \mid \boldsymbol{\epsilon}, \mathbf{t} = t\} \\
&= \int q(\hat{\mathbf{z}} \mid \boldsymbol{\epsilon}, \mathbf{t} = t) q(\mathbf{t} = t \mid \boldsymbol{\epsilon}, h(\mathbf{z}, \boldsymbol{\delta})) dt \quad \{\mathbf{t} = g(\boldsymbol{\epsilon}, h(\mathbf{z}, \boldsymbol{\delta}))\} \qquad (12) \\
&= \int q(\hat{\mathbf{z}} \mid \boldsymbol{\epsilon}, \mathbf{t} = t, h(\mathbf{z}, \boldsymbol{\delta})) q(\mathbf{t} = t \mid \boldsymbol{\epsilon}, h(\mathbf{z}, \boldsymbol{\delta})) dt \quad \{\hat{\mathbf{z}} \perp\!\!\!\perp h(\mathbf{z}, \boldsymbol{\delta}) \mid \boldsymbol{\epsilon}, \mathbf{t} = t\} \\
&= q(\hat{\mathbf{z}} \mid \boldsymbol{\epsilon}, h(\mathbf{z}, \boldsymbol{\delta})) \\
&= q(\hat{\mathbf{z}} \mid h(\mathbf{z}, \boldsymbol{\delta})) \quad \{(\hat{\mathbf{z}}, h(\mathbf{z}, \boldsymbol{\delta})) \perp\!\!\!\perp \boldsymbol{\epsilon}\}
\end{aligned}
$$

Integrating both sides with respect to $q(\boldsymbol{\epsilon} \mid \mathbf{z}, \boldsymbol{\delta})$ we get

$$\int q(\hat{\mathbf{z}} \mid h(\mathbf{z}, \boldsymbol{\delta})) q(\boldsymbol{\epsilon} = \epsilon \mid \mathbf{z}, \boldsymbol{\delta}) d\epsilon = \int q(\hat{\mathbf{z}} \mid \mathbf{z}, \boldsymbol{\epsilon} = \epsilon, \boldsymbol{\delta}) q(\boldsymbol{\epsilon} = \epsilon \mid \mathbf{z}, \boldsymbol{\delta}) d\epsilon = q(\hat{\mathbf{z}} \mid \mathbf{z}, \boldsymbol{\delta}) \qquad (13)$$

Now, the LHS in eq. (13) is

$$\int q(\hat{\mathbf{z}} \mid h(\mathbf{z}, \boldsymbol{\delta})) q(\boldsymbol{\epsilon} = \epsilon \mid \mathbf{z}, \boldsymbol{\delta}) d\epsilon = q(\hat{\mathbf{z}} \mid h(\mathbf{z}, \boldsymbol{\delta})) \implies q(\hat{\mathbf{z}} \mid h(\mathbf{z}, \boldsymbol{\delta})) = q(\hat{\mathbf{z}} \mid \mathbf{z}, \boldsymbol{\delta}).$$

This means

$$q(\hat{\mathbf{z}} \mid \mathbf{z}, \boldsymbol{\epsilon}, \boldsymbol{\delta}) = q(\hat{\mathbf{z}} \mid h(\mathbf{z}, \boldsymbol{\delta})) = q(\hat{\mathbf{z}} \mid \mathbf{z}, \boldsymbol{\delta})$$

Thus $(\hat{\mathbf{z}}, h(\mathbf{z}, \boldsymbol{\delta})) \perp\!\!\!\perp \boldsymbol{\epsilon}$ implies the joint independence $(\hat{\mathbf{z}}, \mathbf{z}, \boldsymbol{\delta}) \perp\!\!\!\perp \boldsymbol{\epsilon}$. □

**Note.**   The proof above shows that we can recover a control function that satisfies ignorability. In this additive setting with finite support however, both the control function and the true confounder violate another fundamental assumption in causal estimation: *positivity*. To see this violation of positivity notice that $p(\mathbf{t} > a + \max_{\epsilon \in \text{supp}(\boldsymbol{\epsilon})} g(\epsilon) \mid h(\mathbf{z}, \boldsymbol{\delta}) = a) = 0$ for any $a$ such that $p(\mathbf{t} > a + \max_{\epsilon \in \text{supp}(\boldsymbol{\epsilon})} g(\epsilon)) > 0$ and $p(h(\mathbf{z}, \boldsymbol{\delta}) = a) > 0$. When positivity is violated, further assumptions are needed to compute causal effects on the whole support of $\mathbf{t}$ in general. Without further assumptions, effects can only be computed on a compact subset of $B \subseteq \text{supp}(\mathbf{t})$ within which positivity holds.

### A.7   From monotonic treatment processes to joint independence

Imbens and Newey [20] explored identification for settings where the outcome process is non-separable but the treatment is a strictly monotonic function of the unobserved confounder. We show that if the reconstruction map $d(\hat{\mathbf{z}}, \boldsymbol{\epsilon})$ reflects this monotonicity condition and $\hat{\mathbf{z}} \perp\!\!\!\perp \boldsymbol{\epsilon}$, the control function is determined by the true confounder and therefore joint independence holds. In VDE, the decoder would be monotonic to reflect this assumption.

**Lemma 2.** *Let $\boldsymbol{\epsilon}$ and $\mathbf{z}$ be the true IV and confounder respectively. Let $\mathbf{z}$ be a continuous scalar.*

1. *Assume that $\mathbf{z}$ has a continuous strictly monotonic CDF. Let the true treatment process be $\mathbf{t} = g(\boldsymbol{\epsilon}, \mathbf{z})$ where $g$ is strictly monotonic in the second argument.*

2. *Let the control function be $\hat{\mathbf{z}} = e(\boldsymbol{\epsilon}, \mathbf{t})$ and let $\hat{\mathbf{z}} \perp\!\!\!\perp \boldsymbol{\epsilon}$. Let reconstruction map be $d$ where $\mathbf{t} = d(\boldsymbol{\epsilon}, \hat{\mathbf{z}})$. Let $e(\cdot, \cdot)$ and $d(\cdot, \cdot)$ be strictly monotonic in the second argument[5].*

3. *Assume that the functions $g, e, d$ are continuous in the second argument and exist for almost every value in the first argument.*

*Then, the control function $\hat{\mathbf{z}}$ can be expressed as a deterministic function of the true confounder $\mathbf{z}$.*

*Proof.*   First, note that $\mathbf{t}$ can be written as a function of $\boldsymbol{\epsilon}$ and a uniform random variable $\mathbf{u}$ using the CDF-inverse trick. Let $H(z) = F(\mathbf{z} \leq z)$. By strict monotonicity and continuity of $H$, $H^{-1}$ exists and $\mathbf{z} = H^{-1}(\mathbf{u})$ for a uniform random variable $\mathbf{u} \perp\!\!\!\perp \boldsymbol{\epsilon}$:

$$\mathbf{t} = g(\boldsymbol{\epsilon}, \mathbf{z}) = g(\boldsymbol{\epsilon}, H^{-1}(\mathbf{u})) = \hat{g}(\boldsymbol{\epsilon}, \mathbf{u}).$$

Note that $H^{-1}$ is strictly monotonic. So, $\hat{g}$ is a strictly monotonic function in the second argument.

Second, due to $\hat{\mathbf{z}} \perp\!\!\!\perp \boldsymbol{\epsilon}$, the conditional CDF of $\hat{\mathbf{z}} \mid \boldsymbol{\epsilon} = \epsilon$ is the same as the marginal CDF as $\hat{\mathbf{z}}$ for almost every value $\epsilon \in \text{supp}(\boldsymbol{\epsilon})$; let's call this CDF $\hat{H}$. By the definition $\hat{\mathbf{z}} = e(\boldsymbol{\epsilon}, \mathbf{t})$ we can express $\hat{\mathbf{z}} = e(\boldsymbol{\epsilon}, \hat{g}(\boldsymbol{\epsilon}, \mathbf{u}))$. Now, $e(\cdot, \cdot), \hat{g}(\cdot, \cdot)$ are both continuous and strictly monotonic in the second argument. So, $\hat{\mathbf{z}}$'s CDF $\hat{H}$ is also strictly monotonic and $\hat{H}^{-1}$ exists and is again strictly monotonic. Therefore, for almost any $\epsilon \in \text{supp}(\boldsymbol{\epsilon})$, we can construct a new uniform random variable by applying $\hat{\mathbf{z}}$'s CDF $\hat{H}$ to $\hat{\mathbf{z}}$:

$$\mathbf{v} = \hat{H}(\hat{\mathbf{z}}) = \hat{H}(e(\boldsymbol{\epsilon}, \hat{g}(\boldsymbol{\epsilon}, \mathbf{u}))).$$

For simplicity, let $\mathbf{v} = J(\boldsymbol{\epsilon}, \mathbf{u})$. Note $J(\cdot, u)$ is strictly monotonic in $u$ by strict monotonicity of $\hat{H}, \hat{g}$ in their second arguments. So, we can write $\mathbf{u}$'s CDF in terms of $\mathbf{v}$'s CDF:

$$a = P(\mathbf{u} < a) = P(\mathbf{v} < J(\epsilon, a)) = J(\epsilon, a).$$

This means that $J(\epsilon, a)$ is an identity function for almost any $\epsilon \in \text{supp}(\boldsymbol{\epsilon})$.

Finally, we can write $\hat{\mathbf{z}}$ as a function of $\mathbf{z}$ for almost any $\epsilon \in \text{supp}(\boldsymbol{\epsilon})$, completing the proof:

$$\hat{\mathbf{z}} = \hat{H}^{-1}(J(\epsilon, H(\mathbf{z}))) = \hat{H}^{-1}(H(\mathbf{z}))$$

□

## A.8 Comparion against other identification results

Imbens and Newey [20] consider non-separable outcome processes, i.e. $\mathbf{y} = f(\mathbf{t}, \mathbf{z})$ and construct control functions by assuming that 1) treatment is a strictly monotonic function of the confounder, 3) the confounder is continuous with a strictly monotonic CDF, and 2) positivity holds for $\mathbf{t}$ with respect to $\mathbf{z}$. These assumptions also lead to identification with general control functions due to the following: a) the positivity assumption is equivalent to the strong IV assumption and b) like additivity, the strict monotonicity assumption reflected in the reconstruction map $d(\hat{\mathbf{z}}, \epsilon)$ as a function of $\hat{\mathbf{z}}$ helps guarantee joint independence; see appendix A.7.

2SLS requires the outcome process to be additive, $\mathbf{y} = f(\mathbf{t}) + \mathbf{z}$. Further, 2SLS needs a "completeness" property: the causal effect function and IV are correlated [16]. While joint independence may not be guaranteed by the completeness condition, it can be guaranteed in certain settings that violate completeness. An example is multiplicative treatment $\mathbf{t} = \mathbf{z} * \epsilon$ with $\mathbf{z} \sim \mathcal{N}(0, 1)$ and a linear outcome; 2SLS fails because $\mathbb{E}[\mathbf{t}\epsilon] = 0$. When joint independence can be guaranteed and the IV is strong, identification with general control functions does not require structural restrictions like additivity of the outcome process that both 2SLS and CFN rely on.

## A.9 Estimation error bounds

We give an example of how violations in reconstruction and independence affect errors in effects.

### A.9.1 GCFN's estimation error in additive treatment process

**Theorem 2.** *Assume an additive treatment process* $\mathbf{t} = \mathbf{z} + g(\epsilon)$ *where* $g$ *is an* $L_g$-*Lipschitz function, and* $\mathbb{E}_{F(\mathbf{z})}\mathbf{z} = 0$. *Let* $\mathbb{E}[\mathbf{y} \mid \mathbf{t} = t, \mathbf{z} = z] = f(t, z)$ *be an* $L$-*Lipschitz function in* $z$ *for any* $t$. *Further,*

1. *let reconstruction error be non-zero but bounded* $\mathbb{E}_{q(\mathbf{t},\hat{\mathbf{z}},\epsilon)}(\mathbf{t} - \hat{\mathbf{z}} - g'(\epsilon))^2 \leq \delta$. *Assume that* $g'$ *is also* $L_g$-*Lipschitz. Further, let* $\mathbb{E}_{q(\hat{\mathbf{z}})}\hat{\mathbf{z}} = 0$, *and* $\mathbb{E}_{q(\hat{\mathbf{z}})}|\hat{\mathbf{z}}| < \infty$.
2. *Assume* $\epsilon \not\perp\!\!\!\perp \hat{\mathbf{z}}$ *and let the dependence be bounded:* $\max_{\hat{z}} \mathcal{W}_1 \left( q(\epsilon \mid \hat{\mathbf{z}} = \hat{z}) \parallel F(\epsilon) \right) \leq \gamma$.

*With the estimated and true causal effects as* $\hat{\tau}(t) = \mathbb{E}_{\hat{\mathbf{z}}} f(t, \hat{\mathbf{z}})$ *and* $\tau(t) = E_{\mathbf{z}} f(t, \mathbf{z})$ *respectively,*

$$\mathbb{E}_{F(\mathbf{t})}|\hat{\tau}(\mathbf{t}) - \tau(\mathbf{t})| \leq L\sqrt{\delta + 4\gamma L_g \mathbb{E}_{q(\hat{\mathbf{z}})}|\hat{\mathbf{z}}|}.$$

*Proof.* Recall the true data distribution is $F(\mathbf{t}, \mathbf{z}, \epsilon)$ such that $\mathbf{z} \perp\!\!\!\perp \epsilon$ and the implied joint $q(\hat{\mathbf{z}}, \mathbf{t}, \mathbf{z}, \epsilon) = q(\hat{\mathbf{z}} \mid \mathbf{t}, \epsilon) F(\mathbf{t}, \mathbf{z}, \epsilon)$. For any $L$-Lipschitz function $\ell(\epsilon)$:

$$
\begin{aligned}
|\mathbb{E}_{q(\epsilon,\hat{\mathbf{z}})}\ell(\epsilon)\hat{\mathbf{z}}| &= |\mathbb{E}_{q(\hat{\mathbf{z}})} \left( \hat{\mathbf{z}} \mathbb{E}_{q(\epsilon \mid \hat{\mathbf{z}})}\ell(\epsilon) \right) - \left( \mathbb{E}_{q(\hat{\mathbf{z}})F(\epsilon)}\hat{\mathbf{z}}\ell(\epsilon) \right)| \quad \{\mathbb{E}_{q(\hat{\mathbf{z}})}\hat{\mathbf{z}} = 0\} \\
&= \left| \mathbb{E}_{q(\hat{\mathbf{z}})} \left( \hat{\mathbf{z}} \left( \mathbb{E}_{q(\epsilon \mid \hat{\mathbf{z}})}\ell(\epsilon) - \mathbb{E}_{F(\epsilon)}\ell(\epsilon) \right) \right) \right| \\
&\leq \mathbb{E}_{q(\hat{\mathbf{z}})}|\hat{\mathbf{z}}| \left| \mathbb{E}_{q(\epsilon \mid \hat{\mathbf{z}})}\ell(\epsilon) - \mathbb{E}_{F(\epsilon)}\ell(\epsilon) \right| \\
&\leq L\mathbb{E}_{q(\hat{\mathbf{z}})}|\hat{\mathbf{z}}|\mathcal{W}_1 \left( q(\epsilon \mid \hat{\mathbf{z}}) \parallel F(\epsilon) \right) \\
&\leq \gamma L\mathbb{E}_{q(\hat{\mathbf{z}})}|\hat{\mathbf{z}}|.
\end{aligned}
\tag{14}
$$

Using the definition of the additive treatment process and the reconstruction error bound, $\mathbb{E}_{q(\mathbf{z},\hat{\mathbf{z}},\epsilon)}(\mathbf{z} + g(\epsilon) - \hat{\mathbf{z}} - g'(\epsilon))^2 = \mathbb{E}_{q(\mathbf{t},\hat{\mathbf{z}},\epsilon)}(\mathbf{t} - \hat{\mathbf{z}} - g'(\epsilon))^2 \leq \delta$. Now, we can bound error in $\hat{\mathbf{z}}$ approximating $\mathbf{z}$

$$
\begin{aligned}
\delta &\geq \mathbb{E}_{q(\mathbf{z},\hat{\mathbf{z}},\epsilon)}(\mathbf{z} - \hat{\mathbf{z}} + g(\epsilon) - g'(\epsilon))^2 \\
&= \mathbb{E}_{q(\mathbf{z},\hat{\mathbf{z}})}(\mathbf{z} - \hat{\mathbf{z}})^2 + \mathbb{E}_{F(\epsilon)}(g(\epsilon) - g'(\epsilon))^2 + 2\mathbb{E}_{q(\mathbf{z},\hat{\mathbf{z}},\epsilon)}(\mathbf{z} - \hat{\mathbf{z}})(g(\epsilon) - g'(\epsilon)) \\
&\geq \mathbb{E}_{q(\mathbf{z},\hat{\mathbf{z}})}(\mathbf{z} - \hat{\mathbf{z}})^2 + 2\mathbb{E}_{q(\mathbf{z},\hat{\mathbf{z}},\epsilon)}(\mathbf{z} - \hat{\mathbf{z}})(g(\epsilon) - g'(\epsilon)) \\
&= \mathbb{E}_{q(\mathbf{z},\hat{\mathbf{z}})}(\mathbf{z} - \hat{\mathbf{z}})^2 + 2\mathbb{E}_{F(\mathbf{z})F(\epsilon)}\mathbf{z}(g(\epsilon) - g'(\epsilon)) - 2\mathbb{E}_{q(\hat{\mathbf{z}},\epsilon)}\hat{\mathbf{z}}(g(\epsilon) - g'(\epsilon)) \quad \{\mathbf{z} \perp\!\!\!\perp \epsilon\} \\
&= \mathbb{E}_{q(\mathbf{z},\hat{\mathbf{z}})}(\mathbf{z} - \hat{\mathbf{z}})^2 + 0 - 2\mathbb{E}_{q(\hat{\mathbf{z}},\epsilon)}\hat{\mathbf{z}}(g(\epsilon) - g'(\epsilon)) \quad \{\mathbb{E}_{F(\mathbf{z})}\mathbf{z} = 0\} \\
&\geq \mathbb{E}_{q(\mathbf{z},\hat{\mathbf{z}})}(\mathbf{z} - \hat{\mathbf{z}})^2 - 4\gamma L_g \mathbb{E}_{q(\hat{\mathbf{z}})}|\hat{\mathbf{z}}| \quad \{g(\epsilon) - g'(\epsilon) \text{ is } 2L_g\text{-Lipschitz}\}
\end{aligned}
$$

Thus, $\mathbb{E}_{q(\mathbf{z},\hat{\mathbf{z}})}(\mathbf{z} - \hat{\mathbf{z}})^2 \leq \delta + 4\gamma L_g \mathbb{E}_{q(\hat{\mathbf{z}})}|\hat{\mathbf{z}}|$. We bound the absolute error in causal effect due to using $\hat{\mathbf{z}}$ instead of $\mathbf{z}$

$$
\begin{aligned}
\mathbb{E}_{\mathbf{t}}|\hat{\tau}(\mathbf{t}) - \tau(\mathbf{t})| &= \mathbb{E}_{\mathbf{t}}|\mathbb{E}_{q(\hat{\mathbf{z}})} f(\mathbf{t}, \hat{\mathbf{z}}) - \mathbb{E}_{F(\mathbf{z})} f(\mathbf{t}, \mathbf{z})| \\
&= \mathbb{E}_{\mathbf{t}}|\mathbb{E}_{q(\hat{\mathbf{z}}, \mathbf{z})} \left( f(\mathbf{t}, \hat{\mathbf{z}}) - f(\mathbf{t}, \mathbf{z}) \right)| \\
&\leq \mathbb{E}_{\mathbf{t}} \mathbb{E}_{q(\hat{\mathbf{z}}, \mathbf{z})} |f(\mathbf{t}, \hat{\mathbf{z}}) - f(\mathbf{t}, \mathbf{z})| \\
&\leq \mathbb{E}_{\mathbf{t}} L \mathbb{E}_{q(\hat{\mathbf{z}}, \mathbf{z})}|\hat{\mathbf{z}} - \mathbf{z}| \\
&\leq L \mathbb{E}_{\mathbf{t}} \sqrt{\mathbb{E}_{q(\hat{\mathbf{z}}, \mathbf{z})} (\hat{\mathbf{z}} - \mathbf{z})^2} \quad \text{(Cauchy-Schwarz)} \\
&\leq L \sqrt{\delta + 4\gamma L_g \mathbb{E}_{q(\hat{\mathbf{z}})}|\hat{\mathbf{z}}|}
\end{aligned}
\tag{15}
$$

When sample size goes to $\infty$, we can guarantee that reconstruction becomes perfect, meaning that $\delta \to 0$, and that $\hat{\mathbf{z}} \perp\!\!\!\perp \epsilon$ holds, meaning that $\gamma \to 0$. Then, this error bound on effects becomes 0.

$\square$

### A.9.2 Bounding effect estimation error

Here, we show that if positivity holds for $\mathbf{t}$ w.r.t. $\mathbf{z}$, and $\mathbf{t}$ w.r.t. $\hat{\mathbf{z}}$, the residual confounding given $\hat{\mathbf{z}}$, i.e. $\mathbf{I}(\mathbf{z}; \mathbf{t} \mid \hat{\mathbf{z}})$, controls the expected absolute error in effects if $q(\mathbf{z} \mid \hat{\mathbf{z}})$ is sufficiently concentrated.

**Theorem 3.** *Let $F(\mathbf{y}, \mathbf{t}, \mathbf{z}, \epsilon)$ be the true data distribution. Let $q(\mathbf{y}, \mathbf{t}, \hat{\mathbf{z}}) = \int F(\mathbf{y}, \mathbf{t}, \mathbf{z} = z, \epsilon) q(\hat{\mathbf{z}} \mid \mathbf{t}, \epsilon) dz d\epsilon$. With $\tau(t^*)$ and $\hat{\tau}(t^*)$ as the true and estimated causal effect of $do(\mathbf{t} = t^*)$ respectively, let $\omega(t^*) = |\hat{\tau}(t^*) - \tau(t^*)|$ be the error. We assume the following.*

1. *Assume that $\mathbf{t}$ satisfies positivity with respect to $\mathbf{z}$, and $\mathbf{t}$ satisfies positivity with respect to $\hat{\mathbf{z}}$.*
2. *Let $\mathbb{E}[\mathbf{y} \mid \mathbf{t} = t, \mathbf{z} = z]$ where $\mathbb{E}$ is w.r.t. $F$, be an $L_t$-Lipschitz function of $z$, for any $t$.*
3. *Let $L := \sup_t L_t$. Let $W := \sup_{t, \hat{z}} {}^{F(\mathbf{t}=t)}\!/_{q(\mathbf{t}=t \mid \hat{\mathbf{z}}=\hat{z})}$.*
4. *Assume $q(\mathbf{z} \mid \hat{\mathbf{z}})$ satisfies the transportation inequality $T_1(\sigma^2/2)$ [29].*

*Then, the expected absolute error in effects is bounded as:* $\quad \mathbb{E}_{F(\mathbf{t})}\omega(\mathbf{t}) \leq \sigma L \sqrt{W \mathbf{I}(\mathbf{z}; \mathbf{t} \mid \hat{\mathbf{z}})}$.

*Proof.* (of theorem 3) Positivity of $\mathbf{t}$ w.r.t. $\mathbf{z}$ implies the conditional expectation $\mathbb{E}[\mathbf{y} \mid \mathbf{z} = z, \mathbf{t} = t^*]$ exists for all $z \in \text{supp}(F(\mathbf{z})), t^* \in \text{supp}(\mathbf{t})$. Positivity of $\mathbf{t}$ w.r.t. $\hat{\mathbf{z}}$ implies the conditional expectation $\mathbb{E}[\mathbf{y} \mid \hat{\mathbf{z}} = \hat{z}, \mathbf{t} = t^*]$ exists for all $\hat{z} \in \text{supp}(F(\hat{\mathbf{z}})), t^* \in \text{supp}(\mathbf{t})$. We begin by expanding the expectation $\mathbb{E}[\mathbf{y} \mid \hat{\mathbf{z}} = \hat{z}, \mathbf{t} = t^*]$ as an integral over the conditional $F(\mathbf{y} \mid \mathbf{z}, \mathbf{t}, \hat{\mathbf{z}}) q(\mathbf{z} \mid \mathbf{t}, \hat{\mathbf{z}})$.

$$
\begin{aligned}
\mathbb{E}[y \mid \hat{\mathbf{z}} = \hat{z}, \mathbf{t} = t^*] &= \int \mathbb{E}\left[\mathbf{y} \mid \mathbf{z} = z, \mathbf{t} = t^*, \hat{\mathbf{z}} = \hat{z}\right] q(\mathbf{z} = z \mid \mathbf{t} = t^*, \hat{\mathbf{z}} = \hat{z}) dz \\
&= \int \mathbb{E}\left[\mathbf{y} \mid \mathbf{z} = z, \mathbf{t} = t^*\right] q(\mathbf{z} = z \mid \mathbf{t} = t^*, \hat{\mathbf{z}} = \hat{z}) dz \quad \{\text{by } \mathbf{y} \perp\!\!\!\perp \hat{\mathbf{z}} \mid \mathbf{t}, \mathbf{z}\},
\end{aligned}
$$

where the inner expectation is with respect to the conditional distribution $F(\mathbf{y} \mid \mathbf{t}, \mathbf{z})$. Now, we prove the bound on $\omega(t^*)$ by expanding the true and estimated effects as expectations over $\mathbf{z}$:

$$
\begin{aligned}
\omega(t^*) &= |\tau(t^*) - \hat{\tau}(t^*)| \\
&= \left| \int \left[ F(\mathbf{z} = z) - \mathbb{E}_{q(\hat{\mathbf{z}})} q(\mathbf{z} = z \mid \mathbf{t} = t^*, \hat{\mathbf{z}}) \right] \mathbb{E}\left[\mathbf{y} \mid \mathbf{z} = z, \mathbf{t} = t^*\right] dz \right| \\
&= L_{t^*} \left| \int \left[ F(\mathbf{z} = z) - \mathbb{E}_{q(\hat{\mathbf{z}})} q(\mathbf{z} = z \mid \mathbf{t} = t^*, \hat{\mathbf{z}}) \right] \frac{\mathbb{E}\left[\mathbf{y} \mid \mathbf{z} = z, \mathbf{t} = t^*\right]}{L_t^*} dz \right| \\
&= L_{t^*} \left| \int \left[ \mathbb{E}_{q(\hat{\mathbf{z}})} \left( q(\mathbf{z} = z \mid \hat{\mathbf{z}}) - q(\mathbf{z} = z \mid \mathbf{t} = t^*, \hat{\mathbf{z}}) \right) \right] \frac{\mathbb{E}\left[\mathbf{y} \mid \mathbf{z} = z, \mathbf{t} = t^*\right]}{L_t^*} dz \right| \\
&\leq L_{t^*} \mathbb{E}_{q(\hat{\mathbf{z}})} \left| \int \left[ (q(\mathbf{z} = z \mid \hat{\mathbf{z}}) - q(\mathbf{z} = z \mid \mathbf{t} = t^*, \hat{\mathbf{z}})) \right] \frac{\mathbb{E}\left[\mathbf{y} \mid \mathbf{z} = z, \mathbf{t} = t^*\right]}{L_t^*} dz \right| \\
&\leq L_{t^*} \mathbb{E}_{q(\hat{\mathbf{z}})} \mathcal{W}_1 \left( q(\mathbf{z} \mid \mathbf{t} = t^*, \hat{\mathbf{z}}) \parallel q(\mathbf{z} \mid \hat{\mathbf{z}}) \right) \\
&\qquad \{{}^{\mathbb{E}[\mathbf{y} \mid \mathbf{z}=z, \mathbf{t}=t^*]}\!/_{L_{t^*}} \text{ is 1-Lipschitz}\} \\
&\leq L_{t^*} \mathbb{E}_{q(\hat{\mathbf{z}})} \sigma \sqrt{\text{KL} \left( q(\mathbf{z} \mid \mathbf{t} = t^*, \hat{\mathbf{z}}) \parallel q(\mathbf{z} \mid \hat{\mathbf{z}}) \right)}
\end{aligned}
$$

$$\leq L_{t^*}\sigma\sqrt{\mathbb{E}_{q(\hat{\mathbf{z}})}\mathrm{KL}\left(q(\mathbf{z}\mid\mathbf{t}=t^*,\hat{\mathbf{z}})\parallel q(\mathbf{z}\mid\hat{\mathbf{z}})\right)}\quad\{\text{by Cauchy Schwarz}\},$$

where the $\mathcal{W}_1$ term was bounded by **KL** by the assumption that $q(\mathbf{z}\mid\hat{\mathbf{z}})$ satisfies the transportation inequality $T_1(\sigma^2/2)$ [29]. Using $L=\sup_t L_t$ and $W={}^{F(\mathbf{t}=t)}/_{q(\mathbf{t}=t\mid\hat{\mathbf{z}}=\hat{z})}$, we can bound the average absolute error

$$\mathbb{E}_{F(\mathbf{t})}\omega(\mathbf{t})\leq\sigma\mathbb{E}_{F(\mathbf{t})}L_{\mathbf{t}}\sqrt{\mathbb{E}_{q(\hat{\mathbf{z}})}\mathrm{KL}\left(q(\mathbf{z}\mid\mathbf{t},\hat{\mathbf{z}})\parallel q(\mathbf{z}\mid\hat{\mathbf{z}})\right)}$$

$$\leq\sigma L\sqrt{\mathbb{E}_{q(\hat{\mathbf{z}})}\mathbb{E}_{F(\mathbf{t})}\mathrm{KL}\left(q(\mathbf{z}\mid\mathbf{t},\hat{\mathbf{z}})\parallel q(\mathbf{z}\mid\hat{\mathbf{z}})\right)}\quad\{\text{by Cauchy Schwarz}\}$$

$$=\sigma L\sqrt{\mathbb{E}_{q(\hat{\mathbf{z}})}\mathbb{E}_{q(\mathbf{t}\mid\hat{\mathbf{z}})}\frac{F(\mathbf{t})}{q(\mathbf{t}\mid\hat{\mathbf{z}})}\mathrm{KL}\left(q(\mathbf{z}\mid\mathbf{t},\hat{\mathbf{z}})\parallel q(\mathbf{z}\mid\hat{\mathbf{z}})\right)}$$

$$\leq\sigma L\sqrt{W}\sqrt{\mathbb{E}_{q(\hat{\mathbf{z}})}\mathbb{E}_{q(\mathbf{t}\mid\hat{\mathbf{z}})}\mathrm{KL}\left(q(\mathbf{z}\mid\mathbf{t},\hat{\mathbf{z}})\parallel q(\mathbf{z}\mid\hat{\mathbf{z}})\right)}$$

$$=\sigma L\sqrt{W}\sqrt{\mathbf{I}(\mathbf{t};\mathbf{z}\mid\hat{\mathbf{z}})}$$

$$\square$$

### A.10   Estimation with the Two-stage least-squares method

We first describe the general version of two-stage least-squares method (2SLS). Let the outcome, treatment and IV be $\mathbf{y},\mathbf{t}',\epsilon$ respectively and the true data distribution be $p(\mathbf{t}',\mathbf{y},\epsilon)$.

1. In the first-stage, 2SLS learns the distribution $q(\mathbf{t}\mid\epsilon)$. Given some class of distributions $Q$, the first-stage can be framed as a maximum-likelihood problem:

$$q=\arg\max_{q'\in Q}\mathbb{E}_{p(\mathbf{t}',\epsilon)}\log q'(\mathbf{t}'\mid\epsilon)$$

   In our setup, $\mathbf{t}$ is the *synthetic treatment* sampled from the conditional distribution $q$ estimated in the first stage.

2. In the second-stage, 2SLS learns the conditional distribution of the outcome $\mathbf{y}$ given the *synthetic treatment* $\mathbf{t}$ sampled from the conditional $q(\mathbf{t}\mid\epsilon)$ from the first stage. Given some class of distributions $G$, 2SLS's second-stage can be framed as a maximum-likelihood problem:

$$g=\arg\max_{g'\in G}\mathbb{E}_{p(\mathbf{y},\epsilon)}\mathbb{E}_{q(\mathbf{t}\mid\epsilon)}\log g'(\mathbf{y}\mid\mathbf{t}).$$

   The causal effect estimate is then computed as: $f^*(t)=\mathbb{E}_{g(\mathbf{y}\mid\mathbf{t}=t)}[\mathbf{y}]$.

Typically in settings with continuous $\mathbf{y},\mathbf{t}$, both stages of 2SLS are framed and implemented as least-squares regressions instead of maximum-likelihood problems. See Kelejian [21] for an overview of classical vs. Bayesian two-stage least-squares methods.

In this section, we derive an alternate expression for 2SLS's causal effect estimate $f^*(t)$. Recall that $\mathbf{t}$ is the *synthetic treatment* sampled from the conditional distribution $q$ estimated in the first stage. We assume that both stages of 2SLS are perfectly solved. Note that $\mathbf{t}$ is independently sampled conditioned on $\epsilon$. This imposes the following conditional independencies:

$$\mathbf{y}\perp\!\!\!\perp\mathbf{t}\mid\epsilon,\mathbf{t}'\quad and\quad\mathbf{t}'\perp\!\!\!\perp\mathbf{t}\mid\epsilon.$$

We marginalize out $\mathbf{t}',\epsilon$ from the joint $q(\mathbf{y},\mathbf{t},\mathbf{t}',\epsilon)$ to get the dependence of $\mathbf{y}$ on $\mathbf{t}$:

$$f^*(t)=\mathbb{E}[\mathbf{y}=y\mid\mathbf{t}=t]$$

$$=\int_{t',\epsilon}yq(\mathbf{y}=y,\mathbf{t}'=t',\epsilon=\epsilon\mid\mathbf{t}=t)d\epsilon dydt'$$

$$=\int_{t',\epsilon}yp(\mathbf{y}=y\mid\mathbf{t}'=t',\epsilon=\epsilon,\mathbf{t}=t)q(\epsilon=\epsilon\mid\mathbf{t}=t)p(\mathbf{t}=t'\mid\epsilon=\epsilon,\mathbf{t}=t)d\epsilon dydt'$$

$$=\int_{t',\epsilon}yp(\mathbf{y}=y\mid\mathbf{t}'=t',\epsilon=\epsilon)q(\epsilon=\epsilon\mid\mathbf{t}=t)p(\mathbf{t}'=t'\mid\epsilon=\epsilon)d\epsilon dydt'$$

$$\{\text{by }\mathbf{t}'\perp\!\!\!\perp\mathbf{t}\mid\epsilon,\ \mathbf{t}\perp\!\!\!\perp\mathbf{y}\mid\mathbf{t},\epsilon\},$$

$$(16)$$

which yields

$$f^*(t) = \int y p(\mathbf{y} = y \mid \mathbf{t}' = t', \boldsymbol{\epsilon} = \epsilon) q(\boldsymbol{\epsilon} = \epsilon \mid \mathbf{t} = t) p(\mathbf{t}' = t' \mid \boldsymbol{\epsilon} = \epsilon) d\epsilon dy dt'$$

$$= \mathbb{E}_{q(\boldsymbol{\epsilon} \mid \mathbf{t}=t)} \mathbb{E}_{p(\mathbf{t}' \mid \boldsymbol{\epsilon})} \mathbb{E}[\mathbf{y} \mid \mathbf{t}', \boldsymbol{\epsilon}]. \quad (17)$$

This shows that the effect estimated by 2SLS can be rewritten as

$$f^*(t) = \mathbb{E}[\mathbf{y} \mid \mathbf{t} = t] = \mathbb{E}_{q(\boldsymbol{\epsilon} \mid \mathbf{t}=t)} \mathbb{E}_{p(\mathbf{t}' \mid \boldsymbol{\epsilon})} \mathbb{E}[\mathbf{y} \mid \mathbf{t}', \boldsymbol{\epsilon}]$$

With this, we show that 2SLS's estimation is biased when the outcome process might have multiplicative interactions between treatment and confounders. Consider this data generation:

$$\boldsymbol{\epsilon}, \mathbf{z} \sim \mathcal{N}(0,1), \ \mathbf{t} = \boldsymbol{\epsilon} + \mathbf{z}, \ \mathbf{y} = \mathbf{t} + \mathbf{t}^2 \mathbf{z}.$$

Let $p(\mathbf{t} \mid \boldsymbol{\epsilon})$ be the learned conditional treatment distribution from a perfectly solved first-stage. We use the reverse conditional $p(\boldsymbol{\epsilon} \mid \mathbf{t})$. 2SLS's causal effect estimate can be rewritten as $f(t) = \mathbb{E}_{q(\boldsymbol{\epsilon} \mid \mathbf{t}=t)} \mathbb{E}_{p(\mathbf{t}' \mid \boldsymbol{\epsilon})} \mathbb{E}[\mathbf{y} \mid \mathbf{t}', \boldsymbol{\epsilon}]$. The true causal effect is $f(t) = \mathbb{E}_{p(\mathbf{z})}[\mathbf{t} + \mathbf{t}^2 \mathbf{z} \mid \mathrm{do}(\mathbf{t} = t)] = t$. Note that $\mathbb{E}[\boldsymbol{\epsilon} \mid \mathbf{t} = t] = \mathbb{E}_{\mathbf{z} \sim \mathcal{N}(0,1)}[t - \mathbf{z}] = t$. The 2SLS-estimate is $3t \neq t = f(t)$:

$$f^*(t) = \mathbb{E}_{q(\boldsymbol{\epsilon} \mid \mathbf{t}=t)} \mathbb{E}_{p(\mathbf{t}' \mid \boldsymbol{\epsilon})} \mathbb{E}[\mathbf{y} \mid \mathbf{t}', \boldsymbol{\epsilon}]$$

$$= \mathbb{E}_{q(\boldsymbol{\epsilon} \mid \mathbf{t}=t)} \mathbb{E}_{p(z)} \mathbb{E}[\mathbf{y} \mid \mathbf{t}' = \mathbf{z} + \boldsymbol{\epsilon}, \boldsymbol{\epsilon}]$$

$$= \mathbb{E}_{q(\boldsymbol{\epsilon} \mid \mathbf{t}=t)} \mathbb{E}_{p(z)}[\boldsymbol{\epsilon} + \mathbf{z} + (\boldsymbol{\epsilon} + \mathbf{z})^2 \mathbf{z}] = 3t$$

This shows 2SLS needs to assume properties of the true outcome and treatment processes.

### A.11 The DeepIV objective

DeepIV [18] extends the two-stage least-squares method to use neural networks in both stages of treatment and outcome estimation. For simplicity, we ignore the covariates $\mathbf{x}$. The first stage of DeepIV estimates the conditional density of treatment given the IV. Assuming the first-stage of DeepIV is solved and we have an estimate $p_\theta(\mathbf{t} \mid \boldsymbol{\epsilon})$, the outcome stage of DeepIV solves the following to obtain an estimate $f_\phi(\mathbf{t})$ for the true causal effect $f(t) = \mathbb{E}[\mathbf{y} \mid \mathrm{do}(\mathbf{t} = t)]$:

$$\min_\phi \mathbb{E}_{\mathbf{y}, \boldsymbol{\epsilon}}[\mathbf{y} - \mathbb{E}_{p_\theta(\mathbf{t} \mid \boldsymbol{\epsilon})} f_\phi(\mathbf{t})]^2. \quad (18)$$

This optimization eq. (18) has a subtle issue. We will show that there exist different functions that solve the optimization problem, thereby resulting in different treatment-effect estimates. Assume that the first stage was solved with $\mathbf{t} \sim p(\mathbf{t} \mid \boldsymbol{\epsilon})$. The trouble lies in the fact that eq. (18) averages the function $f_\phi(\mathbf{t})$ over the distribution $p(\mathbf{t} \mid \boldsymbol{\epsilon})$. If there exists a function $f' \neq 0$ such that $\mathbb{E}_{p(\mathbf{t} \mid \boldsymbol{\epsilon})} f'(\mathbf{t}) = 0$, both $f$ and $f + f'$ solve the optimization problem in Equation (18). As there is no way to separate $f$ from functions like $f + f'$, we face a non-identifiability issue.

We show that multiplicative interactions between $\boldsymbol{\epsilon}, \mathbf{z}$ in the true treatment process is a sufficient condition for such functions $f'$ to exist. Consider the following data generation with no confounding:

$$\boldsymbol{\epsilon}, \mathbf{z} \sim \mathcal{N}(0,1), \ \mathbf{t} = \mathbf{z}\boldsymbol{\epsilon}, \ \mathbf{y} = \mathbf{t}^2.$$

Here the true causal effect is $f(t) = t^2$. We will show that $\mathbb{E}_{p(\mathbf{t} \mid \boldsymbol{\epsilon})} f(\mathbf{t}) = \mathbb{E}_{p(\mathbf{t} \mid \boldsymbol{\epsilon})}(f(\mathbf{t}) + \mathbf{t})$, meaning that both $f(t)$ and $f(t) + t$ solve the optimization problem eq. (18). Notice that $\mathbb{E}[\mathbf{t} \mid \boldsymbol{\epsilon}] = 0$ and therefore

$$\mathbb{E}_{p(\mathbf{t} \mid \boldsymbol{\epsilon})}(f(\mathbf{t}) + \mathbf{t}) = \mathbb{E}[(\mathbf{t}^2 + \mathbf{t}) \mid \boldsymbol{\epsilon}] = \mathbb{E}[\mathbf{t}^2 \mid \boldsymbol{\epsilon}] + \mathbb{E}[\mathbf{t} \mid \boldsymbol{\epsilon}] = \mathbb{E}[\mathbf{t}^2 \mid \boldsymbol{\epsilon}] = \mathbb{E}_{p(\mathbf{t} \mid \boldsymbol{\epsilon})}[f(\mathbf{t})].$$

For any constant $a$, the function $t^2 + at$ also solves the optimization problem in eq. (18). This means that multiple solutions to the DeepIV objective exist that are not the true causal effect.

One potential reason that DeepIV may not run into this non-identifiability issue is that an upper bound of the original proposed objective is solved instead. To compute gradients for the original optimization, two independent expectations are needed, which is not sample-efficient; this is called the double-sample problem. So, [18] optimize an upper bound (via Jensen's):

$$\mathbb{E}_{F(\mathbf{y}, \boldsymbol{\epsilon})}[\mathbf{y} - \mathbb{E}_{p_\theta(\mathbf{t} \mid \boldsymbol{\epsilon})} f_\phi(\mathbf{t})]^2 \leq \mathbb{E}_{F(\mathbf{y}, \boldsymbol{\epsilon})} \mathbb{E}_{p_\theta(\mathbf{t} \mid \boldsymbol{\epsilon})}[\mathbf{y} - f_\phi(\mathbf{t})]^2. \quad (19)$$

The RHS above is a log-likelihood problem with a Gaussian likelihood. A general form of this is $\mathbb{E}_{F(\mathbf{y}, \boldsymbol{\epsilon})} \mathbb{E}_{p_\theta(\mathbf{t} \mid \boldsymbol{\epsilon})} \log p_\phi(\mathbf{y} \mid \mathbf{t})$; where $p_\phi$ is supposed to model the distribution of the outcome under $\mathrm{do}(\mathbf{t})$. Finally, as DeepIV is based on 2SLS, DeepIV assumes an additive outcome process to avoid the issues in the previous section.

**DeepIV under multiplicative treatment processes**  We show here that the upper bound that DeepIV minimizes can also produce biased effect estimates when the true treatment process is multiplicative. The upper bound that DeepIV optimizes is:

$$\arg\min_{f^*} \mathbb{E}_{F(\mathbf{y},\boldsymbol{\epsilon})}\mathbb{E}_{p(\mathbf{t} \mid \boldsymbol{\epsilon})}[\mathbf{y} - f^*(\mathbf{t})]^2 = \arg\min_{f^*} \mathbb{E}_{F(\boldsymbol{\epsilon})}\mathbb{E}_{p(\mathbf{t} \mid \boldsymbol{\epsilon})}\mathbb{E}_{F(\mathbf{y} \mid \boldsymbol{\epsilon})}[\mathbf{y} - f^*(\mathbf{t})]^2$$

Note that we use $F(\mathbf{y} \mid \boldsymbol{\epsilon})$ and not $F(\mathbf{y} \mid \mathbf{t}, \boldsymbol{\epsilon})$ because here $\mathbf{t}$ refers to the synthetic treatment sampled from the conditional distribution $p(\mathbf{t} \mid \boldsymbol{\epsilon})$ learned in the first stage of DeepIV, which means $\mathbf{y} \perp\!\!\!\perp \mathbf{t} \mid \boldsymbol{\epsilon}$. We do a bias-variance decomposition of the expectation and refer to terms that do not depend on $h$ as constants $C$ with respect to the optimization.

$$\mathbb{E}_{F(\boldsymbol{\epsilon})}\mathbb{E}_{p(\mathbf{t} \mid \boldsymbol{\epsilon})}\mathbb{E}_{F(\mathbf{y} \mid \boldsymbol{\epsilon})}[\mathbf{y} - f^*(\mathbf{t})]^2 = \mathbb{E}_{F(\boldsymbol{\epsilon})}\mathbb{E}_{p(\mathbf{t} \mid \boldsymbol{\epsilon})}\mathbb{E}_{F(\mathbf{y} \mid \boldsymbol{\epsilon})}[\mathbb{E}[\mathbf{y} \mid \boldsymbol{\epsilon}] - f^*(\mathbf{t})]^2 + \mathbb{E}_{F(\boldsymbol{\epsilon})}[\sigma^2(\mathbf{y} \mid \boldsymbol{\epsilon})]$$
$$= \mathbb{E}_{p(\mathbf{t})}\mathbb{E}_{p(\boldsymbol{\epsilon} \mid \mathbf{t})}\mathbb{E}_{F(\mathbf{y} \mid \boldsymbol{\epsilon})}[\mathbb{E}[\mathbf{y} \mid \boldsymbol{\epsilon}] - f^*(\mathbf{t})]^2 + C \tag{20}$$

Now consider the generation process $\boldsymbol{\epsilon}, \mathbf{z} \sim \mathcal{N}(0,1)$ with the true treatment and outcome generated as $\mathbf{t} = \boldsymbol{\epsilon}\mathbf{z}$ and $\mathbf{y} = \mathbf{t} + \mathbf{z}$. Note that $E[\mathbf{y} \mid \boldsymbol{\epsilon} = a] = E_{\mathbf{z}}[\mathbf{z} + a\mathbf{z}] = 0$. Therefore the optimization reduces to the following:

$$\arg\min_{f^*} \mathbb{E}_{p(\mathbf{t})}\mathbb{E}_{p(\boldsymbol{\epsilon} \mid \mathbf{t})}\mathbb{E}_{F(\mathbf{y} \mid \boldsymbol{\epsilon})}[0 - f^*(\mathbf{t})]^2 = 0 \neq f(t) = t$$

Thus, DeepIV's relaxed optimization problem also needs assumptions on the true treatment process.

## A.12  Information preserving maps and additional utility constraints

A bijective map is one that maps each element in its domain to a unique element in its range. No information can be lost in this process, resulting in bijective transformations being called information-preserving maps. Information-preserving maps preserve computations that only involve conditioning and expectations; meaning that the causal effect estimate $\mathbb{E}_{\hat{\mathbf{z}}}\mathbb{E}[\mathbf{y} \mid \mathbf{t}, \hat{\mathbf{z}}]$ is preserved. Therefore we can impose additional distributional utility constraints satisfied by bijective transformations of the general control function $\hat{\mathbf{z}}$, without losing the properties of ignorability.

Coupled with flexible over-parametrized modelling, information-preserving maps give us the ability to enforce utility constraints on the latent space of $\hat{\mathbf{z}}$. If there is an outcome-model that works well with data drawn from a normal distribution, one can add an additional term to VDE's objective that is the KL divergence between the distribution of $\hat{\mathbf{z}}$ and a normal distribution. If we wanted information about continuity in $\mathbf{t}$ to be preserved in $\hat{\mathbf{z}}$, we could enforce linear interpolation. Similarly, we could force an constructed $\hat{\mathbf{z}}$ to have a monotonic relation with $\mathbf{t}$. One could enforce multiple constraints from a combination of distances, divergences, ordering and modality constraints. When used correctly, these constraints trade optimization complexity between outcome-stage and VDE.

# B  Experimental Details

In this section, we expand on the details of experiments presented in section 4. In all experiments, the hidden layers in both encoder and decoder networks have 100 units and use ReLU activations. The outcome model is also a 2-hidden-layer neural network with ReLU activations unless specified otherwise. For the simulated data, the hidden layers in the outcome model have 50 hidden units. We optimize VDE and outcome-stage for 100 epochs with Adam; starting with a learning rate of $10^{-2}$ and halving it every 10 epochs if the training error goes up.

## B.1  Selecting $\lambda$

We discuss here why good $\lambda$ (equivalently $\kappa$) can be selected based on the resulting expected outcome likelihood, i.e. the outcome modelling objective, on a heldout validation set.

As VDE's control function is constructed as a function of $(\mathbf{t}, \boldsymbol{\epsilon})$, i.e. $\hat{\mathbf{z}} = e(\mathbf{t}, \boldsymbol{\epsilon})$, it holds that $\mathbf{y} \perp\!\!\!\perp \hat{\mathbf{z}} \mid \boldsymbol{\epsilon}, \mathbf{t}$. So, predicting $\mathbf{y}$ from $(\hat{\mathbf{z}}, \mathbf{t})$, as in GCFN, cannot be better than predicting $\mathbf{y}$ from $(\mathbf{t}, \boldsymbol{\epsilon})$:

$$\mathbf{H}(\mathbf{y} \mid \mathbf{t}, \hat{\mathbf{z}}) \geq \mathbf{H}(\mathbf{y} \mid \mathbf{t}, \boldsymbol{\epsilon}, \hat{\mathbf{z}}) = \mathbf{H}(\mathbf{y} \mid \mathbf{t}, \boldsymbol{\epsilon}).$$

The slack in the inequality is $\mathbf{H}(\mathbf{y} \mid \mathbf{t}, \hat{\mathbf{z}}) - \mathbf{H}(\mathbf{y} \mid \mathbf{t}, \boldsymbol{\epsilon}, \hat{\mathbf{z}}) = \mathbf{I}(\mathbf{y}, \boldsymbol{\epsilon} \mid \mathbf{t}, \hat{\mathbf{z}})$ and equality holds when $\mathbf{y} \perp\!\!\!\perp \boldsymbol{\epsilon} \mid \mathbf{t}, \hat{\mathbf{z}}$. This independence holds in general only if both $\mathbf{z} \perp\!\!\!\perp \boldsymbol{\epsilon} \mid \hat{\mathbf{z}}$ and perfect reconstruction hold;

see appendix B.1.1. Thus, in general, the expected outcome likelihood achieves maximum only when both perfect reconstruction and conditional independence are satisfied.

In practice, instead of the unconstrained VDE, we optimize the lower-bound objective in eq. (6) on a finite dataset. Due to local minima or finite-sample error, this lower-bound optimized with a $\kappa$ that is too large may give a $\hat{\mathbf{z}}$ that retains little information about $\mathbf{z}$ so as keep the **KL** small. Similarly, when $\kappa$ is too small, $\hat{\mathbf{z}}$ may memorize $\mathbf{t}$ to keep the reconstruction error small without paying much in the $\kappa \times$ **KL** term. In either case, the resulting $\hat{\mathbf{z}}$ fails to satisfy one of either perfect reconstruction or conditional independence, meaning that $\mathbf{y} \not\!\perp\!\!\!\perp \epsilon \mid \mathbf{t}, \hat{\mathbf{z}}$ in general. Then, as discussed above, the outcome model cannot achieve the maximum possible expected outcome likelihood. This insight suggests the following procedure to select good $\kappa$ based on validation outcome likelihood: [6]:

1. Solve VDE for a collection of $\kappa$ and obtain the control function $\hat{\mathbf{z}}_\kappa$ for each.

2. Regress $\mathbf{y}$ on $\mathbf{t}, \hat{\mathbf{z}}_\kappa$ and evaluate expected outcome likelihood on a heldout validation set. (This heldout set should be different from the one used to tune all other hyperparameters)

3. Select the $\kappa$ that led to the largest validation outcome likelihood; use the corresponding $\hat{\mathbf{z}}_\kappa$ in GCFN's second stage to estimate effects (retrain or use the model from step 2).

### B.1.1 Conditional Independence of outcome and instrument given $\hat{\mathbf{z}}, \mathbf{t}$

By definition, the potential outcome $\mathbf{y_t}$ depends only on $\mathbf{z}$ and for any observed $(\mathbf{t}, \mathbf{y})$, and by consistency, $\mathbf{y} = \mathbf{y_t}$. Therefore $\mathbf{z} \perp\!\!\!\perp \epsilon \mid \hat{\mathbf{z}}, \mathbf{t} \implies \mathbf{y_t} \perp\!\!\!\perp \epsilon \mid \hat{\mathbf{z}}, \mathbf{t} \iff \mathbf{y} \perp\!\!\!\perp \epsilon \mid \hat{\mathbf{z}}, \mathbf{t}$. Under the joint $q(\hat{\mathbf{z}}, \mathbf{t}, \epsilon, \mathbf{z}) = q(\hat{\mathbf{z}} \mid \mathbf{t}, \epsilon) F(\mathbf{t}, \epsilon, \mathbf{z})$, it follows that $\mathbf{z} \perp\!\!\!\perp \epsilon \mid \hat{\mathbf{z}}, \mathbf{t}$ when the reconstruction property and the conditional independence $\mathbf{z} \perp\!\!\!\perp \epsilon \mid \hat{\mathbf{z}}$ hold:

$$
\begin{aligned}
q(\mathbf{z}, \epsilon \mid \hat{\mathbf{z}}, \mathbf{t}) &= q(\mathbf{z} \mid \epsilon, \hat{\mathbf{z}}, \mathbf{t}) q(\epsilon \mid \hat{\mathbf{z}}, \mathbf{t}) \\
&= q(\mathbf{z} \mid \epsilon, \hat{\mathbf{z}}) q(\epsilon \mid \hat{\mathbf{z}}, \mathbf{t}) \quad \{\text{by reconstruction } \mathbf{t} = d(\hat{\mathbf{z}}, \epsilon)\} \\
&= q(\mathbf{z} \mid \hat{\mathbf{z}}) q(\epsilon \mid \hat{\mathbf{z}}, \mathbf{t}) \quad \{\text{by joint independence } \mathbf{z} \perp\!\!\!\perp \epsilon \mid \hat{\mathbf{z}}\} \\
&= q(\mathbf{z} \mid \hat{\mathbf{z}}, \mathbf{t}) q(\epsilon \mid \hat{\mathbf{z}}, \mathbf{t}) \quad \{\text{by joint independence and reconstruction } \mathbf{z} \perp\!\!\!\perp \mathbf{t} \mid \hat{\mathbf{z}}\},
\end{aligned}
\tag{21}
$$

where $\mathbf{z} \perp\!\!\!\perp \mathbf{t} \mid \hat{\mathbf{z}}$ is shown in the proof of theorem 1. If $\mathbf{y_t}$ is an invertible function of $\mathbf{z}$, $\mathbf{y} \perp\!\!\!\perp \epsilon \mid \hat{\mathbf{z}}, \mathbf{t} \implies \mathbf{z} \perp\!\!\!\perp \epsilon \mid \hat{\mathbf{z}}, \mathbf{t}$. Thus, in general, $\mathbf{z} \perp\!\!\!\perp \epsilon \mid \hat{\mathbf{z}}, \mathbf{t}$ is a necessary condition for $\mathbf{y} \perp\!\!\!\perp \epsilon \mid \hat{\mathbf{z}}, \mathbf{t}$.

### B.2 Simulations with Specific Decoder Structure

We used the python package *statsmodels* for 2SLS and our own implementation of CFN. We used the DeepIV package developed by Hartford et al. [18].

**Multiplicative treatment + Additive outcome.** We use the 2SLS function from statsmodels [37] which uses a linear model $\mathbf{t} = \beta \epsilon + \boldsymbol{\eta_t}$ that will correctly predict that $\mathbb{E}[\mathbf{t} \mid \epsilon] = 0$. We optimized the treatment and the response models in DeepIV [18] for a 100 epochs each.

### B.3 GCFN on high-dimensional covariates

Here, we give further details about section 4.3. We give Hartford et al. [18]'s simulation with our notation:

$$
\mathbf{z}, \epsilon \sim \mathcal{N}(0, 1), \quad \mathbf{t} = 25 + (\epsilon + 3)\psi_s + \nu, \quad \mathbf{y} = \mathcal{N}\big(100 + (10 + \mathbf{t})\ell(\mathbf{x})\psi_s - 2\mathbf{t} + 0.5\mathbf{z}, 0.75\big),
$$

where $\psi_s$ is a non-linear function of time $s$, and $\ell(\mathbf{x})$ is the label of the MNIST image. We optimized both VDE and outcome stage with Adam with batch size 500 for 200 epochs beginning at $10^{-2}$ and halving the learning rate when the average loss over 5 epochs increases. We use the outcome model architecture from DeepIV [18] where convolutional layers construct a representation which is concatenated with $\mathbf{t}$ and $s$, before being fed to the fully-connected layers. GCFN's outcome model differs only in that the fully-connected layers take as input the control function $\hat{\mathbf{z}}$, time $s$ and treatment $\mathbf{t}$. The best outcome model was chosen based on validation outcome MSE.

### B.4 GCFN on high-dimensional IV

Here, we give further details about section 4.4. The encoder and additive decoder in VDE are 2-layer networks like in the section 4.1. In this experiment we use a 3 layer outcome model with 50 units in each layer. We used 10,000 samples as in DeepGMM and optimized both VDE and outcome stage with Adam with a batch size 1000 for 100 epochs beginning at a learning rate of $10^{-2}$ and halving it when the average loss over 5 epochs increases. We plot outcome and effect MSE for GCFN for 5 different $\kappa \in \{0.1, 0.2, 0.3, 0.4, 0.5\}$ in fig. 6. Note that low outcome MSE corresponds to

**Figure 6:** GCFN performs on par with DeepGMM on high-dimensional IV experiment specified in DeepGMM [7].

low effect MSE. The plot shows mean and standard deviation of effect MSE of the causal effect for 5 different $\alpha$'s and 10 random seeds. GCFN performs on par or better than all methods given in DeepGMM [7].

### B.5   Additional experiments

The following experiment is done with a structurally unrestricted decoder even though the true treatment process is additive. We compare against CFN to demonstrate that GCFN does not require structural restrictions on the outcome process. Let $\mathcal{N}$ be the normal distribution and $\alpha$ be a parameter to control the confounding strength. We generate

$$\mathbf{z}, \boldsymbol{\epsilon} \sim \mathcal{N}(0,1), \quad \mathbf{t} = (\mathbf{z} + \boldsymbol{\epsilon})/\sqrt{2}, \quad \mathbf{y} \sim \mathcal{N}(\mathbf{t}^2 + \alpha \mathbf{z}^2, 0.1). \qquad (22)$$

The larger the absolute values of $\alpha$, the more the confounding. In economics terminology, the treatment noise and the outcome noise are $\boldsymbol{\eta}_t = \mathbf{z}$ and $\boldsymbol{\eta}_y = \alpha \mathbf{z}^2 + noise$ respectively. The generation process in eq. (22) violates assumption A4 in Guo and Small [16] for CFN: $\mathbb{E}[\boldsymbol{\eta}_y | \boldsymbol{\eta}_t] \propto \boldsymbol{\eta}_t$. GCFN does not require this assumption. We use 5000 samples and a batch size of 500. We discretize the treatment to have 50 categories. Of the 50, 48 categories correspond to equally sized bins in $[-3.5, 3.5]$, with the remaining 2 correspond to values less than $-3.5$ and greater than $3.5$ respectively. We compare against CFN with both stages correctly specified as functions of $\mathbf{t}$ and $\mathbf{z}$.

We find, as expected, that GCFN out-performs CFN. Over 5 runs, for $\alpha = 1$, we obtain an RMSE of $\mathbf{0.3 \pm 0.1}$ while the CFN only manages to obtain an RMSE of $\mathbf{1.5 \pm 0.1}$ despite having the correctly specified model for $\mathbf{t}^2$. For other $\alpha \in \{-2, -1, 2\}$, GCFN was similarly better.

## Footnotes

[3] If $q(\mathbf{z} = z, \boldsymbol{\delta} = \delta \mid \hat{\mathbf{z}}) = 0$ everywhere then no pair $(\mathbf{z} = z, \boldsymbol{\delta} = \delta)$ maps to $\hat{\mathbf{z}}$ and $\hat{\mathbf{z}}$ cannot be observed and we cannot condition on it. But $\hat{\mathbf{z}}$ is constructed explicitly as part of the algorithm, so it's observed. Thus $q(\mathbf{z} = z, \boldsymbol{\delta} = \delta \mid \hat{\mathbf{z}})$ is a valid conditional density.

[4] $P(\mathbf{c} < x) = P(\mathbf{a} + \mathbf{b} < x) + P(1 < \mathbf{a} + \mathbf{b} < 1 + x) = 0.5(x^2 - 1) + 1 - 0.5(1 - x)^2 = x.$

[5]Note that $e(\epsilon, \cdot) = d^{-1}(\epsilon, \cdot)$. Then, monotonicity of $d$ in the second argument implies the same for $e$.

[6]At first glance, one failure case seems to be when $\hat{\mathbf{z}}$ memorizes $\epsilon$ only, leading to $\mathbf{y} \perp\!\!\!\perp \epsilon \mid \mathbf{t}, \hat{\mathbf{z}}$. However, such a $\hat{\mathbf{z}}$ does not help reconstruct $\mathbf{t}$ along with $\epsilon$ while resulting in a large KL $(q(\hat{\mathbf{z}} \mid \mathbf{t}, \epsilon) \parallel q(\hat{\mathbf{z}}))$. This leads to a very sub-optimal objective value in VDE. As we maximize to solve VDE, such failure cases do not occur.