[Reviews · NeurIPS 2020]

Review 1

Summary and Contributions: This paper gives a control function approach for causal inference with an instrumental variable. The proposed method trades typical assumptions made on the outcome process for assumptions made instead on the treatment process; estimation proceeds in two stages, the first via variational autoencoder to estimate the control function and the second via maximum likelihood for outcome modeling. ----- Update post-rebuttal: I was hoping for more comparison against previous identification results than the brief description in Section 2.2. For now it is unclear how the authors expand on this in the updated version. I look forward to seeing how the authors address multiple suggestions to make the contributions and writing more clear.

Strengths: The claims made appear sound. It seems worthwhile to consider settings where usual functional assumptions may hold for the treatment but not outcome.

Weaknesses: The main theoretical contribution is the identification result in Theorem 1, but it is not explained how this relates to standard identification results for control functions, and I cannot immediately see from the result itself (for example it is not clear to me how the assumptions map to those of Guo & Small's 2016 paper). It seems to me that for a new identification result it would be required to compare & contrast with previous results, for example stating which conditions are stronger/weaker, etc. I also found the assumptions required for identification to be difficult to gauge in a real data analysis. The method proposed in Section 2.2 seemed somewhat opaque to me, and no analysis of its properties was given, beyond what I felt was somewhat limited evidence from simulation experiments in Section 3.

Correctness: I did not see errors in the claims or methodology.

Clarity: I found this paper somewhat difficult to follow, despite being quite familiar with the IV literature. I felt it was lacking some useful description and background throughout.

Relation to Prior Work: Beyond the first point made in the Weaknesses section above, it appears the paper cites sufficient relevant work.

Reproducibility: Yes

Additional Feedback:


Review 2

Summary and Contributions: This paper combines ideas from variational inference and the control function approach to instrumental variable estimation to produce a method for instrumental variable estimation that allows weaker constraints on how unobserved confounders affect the response than the common y = f(t,x) + u setting, at the expense of stronger constraints on how the unobserved confounders affect the treatment (though slightly weaker than the original control function setting).

Strengths: This is creative work - it shows how the control function approach, which attempts to estimate the unobserved confounder so that you can use it as a control in the downstream estimation task, can be generalized beyond the additive treatment and additive response model (eqn 1). Theorem 1 summarizes the necessary conditions that allow the authors to weaken these assumptions to additive or monotonic treatment relationships without the constraints on the output model. These conditions are, however, tricky to satisfy, so much of the remainder of the paper focuses on designing a methodology that meets the requirements of the theorem. Theorem 1 requires independence between the control function and the instrument - which the author achieve via their variational decoupling method which minimizes the mutual information between these two variables. Because they're optimizing an upper bound on this quantity & don't appear to be enforcing this constraint exactly (the legrange multiplier appears fixed throughout the optimization), it seems likely that this mutual information minimization is not driven to 0. The authors give bounds on the estimation error that follow from this problem. They also do a decent job of explaining the subtitles around the joint independence requirements that follow from their theorem. On the empirical side - the performance of the method appears to be strong relative to TSLS & DeepIV - particularly on the the cases where the TSLS & DeepIV assumption fail (as one would expect). This is somewhat comforting because, as I discuss below - there are a lot of moving pieces in this model, so it's not clear a priori that it should work.

Weaknesses: - the method needs both 0 mutual information between \hat{z} and \epsilon and perfect reconstruction of the treatment for the conditions of theorem 1 to apply. The estimation error section gives some discussion of bounds that suggests that things don't get too bad, but I really would have liked to see this evaluated experimentally. - some of the sections are clearly written but the overall structure could be better: I would bring lines 187 to 206 before section 2.2 so that section 2.1 deals with all the conditions & assumptions that have to hold for the method to work. Then presenting variational decoupling as your approach to attempting to satisfy those conditions + being explicit about the ways it might fail would make the me less skeptical of the work. In particular, as mentioned above, the error bounds section seems to be an important part of the contribution, but as its currently positioned it feels tacked on... More minor: - I know economists love using the weather as instruments, but a hurricane feels like a poor example - it seems likely that more response variables would also be affected by a hurricane (so it fails the exclusion assumption). - Consider swapping \epsilon and z in your notation... z is widely used as the symbol for an instrument & it's not unusual to use \epsilon as a confounder, so as someone who reads a lot of IV literature I kept getting confused about which variable was which... it felt a bit like you were using y for features and x for the target variable.

Correctness: As far as I can tell the claims are correct.

Clarity: See above - it could definitely be a lot clearer.

Relation to Prior Work: Yes

Reproducibility: Yes

Additional Feedback: Post rebuttal - increased my score to account for the new experiments & proposed changes.


Review 3

Summary and Contributions: UPDATE: I remain at a weak accept. Without seeing the result of improved presentation and clarity I wouldn't increase my score since I already took into account that the presentation might improve. The rebuttal was too brief to really get the message across. I agree with R1's comment that the identification results should be better discussed and compared to related work, and this is reflected in some of my review. Although it does appear that the results have enough novelty. Overall I think this paper could be a strong contribution. Based on the provided information I'm just not convinced enough to give a clear accept. The paper introduces instrumental variable methods for causal inference in models that are non-separable, i.e. where the outcome is not a (weighted) sum of the treatment and confounder. Additionally, other linearity assumptions are relaxed. This task requires specific conditions on the causal graph, which are derived and explained. The paper formulates a variational objective that whose optimization leads to one of these conditions.

Strengths: IV regression is an important area of causal inference, complementing methods that adjust for observed confounders. Existing IV regression methods require strong assumptions on the functional forms generating the treatment and outcome. Here, these are replaced with different and potentially weaker assumptions such as a monotone relationship between the unobserved confounder and the treatment. A sound theoretical analysis shows which independence assumptions are necessary and an elegant constrained optimization (VDE) is developed to satisfy them. Toy experiments show that other methods can fail when their assumptions are violated.

Weaknesses: 1) The experiments are simple toy simulations. I don’t know if this is standard practice in IV regression but it would be desirable to use benchmarks if available. However, the high-dimensional experiment, taken from prior work, is a more challenging/convincing test. The slave export dataset is unconvincing since it only shows that the method gives a similar estimate as another method which requires stronger assumptions but we do not know if that estimate is correct. 2) The methods section and introduction could be written more clearly. Before publishing, I would recommend to give them a thorough re-write. 3) A1 and A2 are not clearly explained. Furthermore, it should be discussed how realistic these assumptions are. Although its formulation should be improved, in my view, theorem 1 is a useful result and leads to a well-justified algorithm VDE. As far as I can see these are novel insights. So I lean towards acceptance.

Correctness: I am not aware of any errors.

Clarity: I would recommend to find a more intuitive structure for section 2 if possible. E.g. some related work has a clear separation between stage 1 and stage 2. Section 3 is clearly structured and easier to follow.

Relation to Prior Work: Yes.

Reproducibility: Yes

Additional Feedback: Kernel instrumental variable regression may be an appropriate additional baseline as it also relaxes linearity assumptions. DeepIV already does this, but the performance of neural nets can be a bit unpredictable. Does VDE work if z is multidimensional, and what would be the necessary assumption - monotonicity/additivity in all components? Typos: “While VDE creates control functions, that are independent of the instrument.”, “when its observed”, “is can put”, “GCFN is also more robust to confounding than DeepIV when the additive outcome process assumption.”, “and guarantees positivity”

[Author Response · NeurIPS 2020]

We thank the reviewers for their detailed feedback. We're glad that the reviewers thought our general identification
theorem is sound and interesting (R1, R2, R3), found our work is creative (R2), and see VDE as an elegant solution to
satisfy the independence conditions required in theorem 1 (R3).

To address clarity concerns, we have adopted the reorganization suggestions of section 2 from R2. This should move the
"identification comparison with other control function methods" closer to theorem 1 which clarifies one of R1's main
concerns by connecting our work to Guo and Small 2016. This would also surface the error analysis of GCFN from
section 2.3 and appendix A.7 that R1 requests and provide the more intuitive structure of section 2 that R3 requests.

**[R1 + R3 : limited experimental evidence | R3, toy experiments]**    We do not know of any established benchmarks
for causal effect estimation with IVs. We briefly discuss our evaluation here. Section 3.1 shows GCFN can handle
assumptions that existing methods like CFN, 2SLS, and DeepIV cannot. Then, as demonstrated below and in section
3.3 in the paper, GCFN performs on par or better than DeepIV and DeepGMM on *high-dimensional* simulated data
from each paper [1, 2] respectively. Thus, GCFN is competitive on data that satisfies the assumptions required by
DeepIV and DeepGMM *while* being applicable to data generated with non-additive outcome processes [1, 2].

**[Additional evaluation of GCFN]**    Due to reviewers' concern about toy experiments, we present further evaluation
of GCFN on simulated data with a high-dimensional IV. We use the data generating process given in DeepGMM [2], a
recent state of art method. We ran GCFN with 10 different random seeds and report results for $\kappa = 0.3$, chosen based
on mean outcome MSE. We report results for DeepGMM and DeepIV as reported in [2]. GCFN performs competitively
with an effect MSE of $\mathbf{0.077 \pm 0.022}$ compared to DeepGMM's $\mathbf{0.07 \pm 0.02}$ and DeepIV's $\mathbf{0.11 \pm 0.00}$. Effect
MSE for $\kappa \in \{0.2, 0.4\}$ were similarly within standard errors of DeepGMM's performance and better than DeepIV [1].

**[R1 + R3, hard to gauge assumptions | realism of assumptions]**    Solving VDE exactly guarantees reconstruction
(A1) and marginal independence $\hat{\mathbf{z}} \perp \epsilon$. A strong IV (A2), intuitively, is one that is able to set treatment to any value
given any fixed confounder value. Joint independence requires assumptions on the treatment process like additivity.
Additivity is common in the economics literature and strong IV is an assumption that can be reasoned about using
domain expertise: for e.g. can college proximity influence a student's decision to go to college regardless of skill?

**[R1, Theorem 1 and traditional identification results]**    To clarify, theorem 1 characterizes control functions that
guarantee identification. We compare GCFN's identification for an additive treatment process to traditional control
function identification from Guo and Small (reference 15 in the paper) in the paragraph "New conditions for effect
identification" in section 2.2. Briefly, a strong IV requires more than the exclusion restriction and relevance properties
of IVs. For this added assumption, GCFN drops the additive outcome process and the additional conditions on noise
that CFN requires. We have expanded this discussion in the paper.

**[R1, Section 2.2 is opaque, no analysis]**    As noted above, we
have clarified our presentation in section 2.2 and A.7. We briefly
summarize GCFN here: 1) GCFN's first stage, VDE, has an ob-
jective where at optimum, control functions meet condition A1 in
theorem 1. Similarly, VDE's decoder leverages treatment process
assumptions where at optimum joint independence (A3) is satisfied
(line 187). 2) GCFN then runs a flexible outcome regression on
the treatment and VDE's control function to estimate effects. As
in section 2.3, we analyse GCFN's effect error in appendix A.7. In
A.7.1, we show how the two components of VDE, reconstruction
error and dependence of $\hat{\mathbf{z}}$ on $\epsilon$, influence effect error in data with
additive treatment processes. We also give a general bound in A.7.2.

**[R2, $I(\hat{\mathbf{z}}, \epsilon) > 0$, reconstruction error, GCFN failure cases]**
We thank the reviewer for this suggestion. We will add discussion
about failures modes of GCFN. We plot the influence of non-zero
information and reconstruction on effect MSE in fig. 1 using models
trained on the MNIST IV data above for different $\kappa$.

Figure 1: Effect estimation MSE of GCFN in the MNIST IV experiment above plotted for the corresponding VDE's reconstruction loss and upper bound on $I(\hat{\mathbf{z}}, \epsilon)$ (as in derivation in A.2); the lighter the color, the better the error. Effect MSE is good and very mildly sensitive to changes in reconstruction or information for values in $0.3 - 0.5$ and $2.5 - 3.5$ respectively. Outside those ranges, effect MSE is more sensitive and is bad when either quantity is large.

**[R3, Real experiment unconvincing]**    The reported effect in Nunn et al. (reference [34] in the paper) is a well-known
result supported by modelling choices informed by domain knowledge. We believe that recovering this effect gives
evidence that GCFN works well even without strong parametric assumptions used by Nunn et al.

**[R3, Kernel IV, Multidimensional z]**    We thank the reviewer for the suggestion. We will include a Kernel IV baseline.
If $\mathbf{t}$ is scalar but $\mathbf{z}$ is multidimensional, an additive treatment process with a scalar $f(\mathbf{z})$ would be $\mathbf{t} = g(\epsilon) + f(\mathbf{z})$. In
this case a scalar $\hat{\mathbf{z}}$ suffices to capture $f(\mathbf{z})$ and satisfy ignorability.

[1]  Jason Hartford, Greg Lewis, Kevin Leyton-Brown, and Matt Taddy. Deep iv: A flexible approach for counterfactual prediction.
In *Proceedings of the 34th International Conference on Machine Learning-Volume 70*, 2017.
[2]  Andrew Bennett, Nathan Kallus, and Tobias Schnabel. Deep generalized method of moments for instrumental variable analysis.
In *Advances in Neural Information Processing Systems*, 2019.


[Meta-Review · NeurIPS 2020]

The reviewers and myself are in agreement that the paper proposes an interesting methodological approach for the instrumental variable regression problem without the additive separability assumption, that is widespread in the literature. The approach also combines interesting ideas from both the econometrics/statistics literature (control functions) and the machine learning literature (autoencoders). The reviewers continue to have concerns on the relation of the identification results to prior literature. The authors tried to address this in the rebuttal and point to a relevant discussion in the paper. However, the short discussion in the paper and in the rebuttal is not sufficient. The authors are therefore strongly encouraged to make the relationship to prior identification theorems in the control function literature and relationship to prior assumptions more concrete, elaborate and technical. Despite this drawback, the paper offers a strong methodological contribution that works well in practice and the theoretical component seems to mostly be a secondary sanity check that the method has theoretical grounding. Given this I am willing to disregard the non-elaborate comparison to the prior identification theorems and recommend that this paper be accepted.